# SCANDL: A Diffusion Model for Generating Synthetic Scanpaths on Texts

**Lena S. Bolliger**[1], **David R. Reich**[2], **Patrick Haller**[1], **Deborah N. Jakobi**[1]
**Paul Prasse**[2], **Lena A. Jäger**[1,2]

[1]Department of Computational Linguistics, University of Zurich, Switzerland
[2]Department of Computer Science, University of Potsdam, Germany
{bolliger,haller,jakobi,jaeger}@cl.uzh.ch
{david.reich,paul.prasse}@uni-potsdam.de

## Abstract

Eye movements in reading play a crucial role in psycholinguistic research studying the cognitive mechanisms underlying human language processing. More recently, the tight coupling between eye movements and cognition has also been leveraged for language-related machine learning tasks such as the interpretability, enhancement, and pre-training of language models, as well as the inference of reader- and text-specific properties. However, scarcity of eye movement data and its unavailability at application time poses a major challenge for this line of research. Initially, this problem was tackled by resorting to cognitive models for synthesizing eye movement data. However, for the sole purpose of generating human-like scanpaths, purely data-driven machine-learning-based methods have proven to be more suitable. Following recent advances in adapting diffusion processes to discrete data, we propose SCANDL, a novel discrete sequence-to-sequence diffusion model that generates synthetic scanpaths on texts. By leveraging pre-trained word representations and jointly embedding both the stimulus text and the fixation sequence, our model captures multi-modal interactions between the two inputs. We evaluate SCANDL within- and across-dataset and demonstrate that it significantly outperforms state-of-the-art scanpath generation methods. Finally, we provide an extensive psycholinguistic analysis that underlines the model's ability to exhibit human-like reading behavior. Our implementation is made available at https://github.com/DiLi-Lab/ScanDL.

## 1 Introduction

As human eye movements during reading provide both insight into the cognitive mechanisms involved in language processing (Rayner, 1998) and information about the key properties of the text (Rayner, 2009), they have been attracting increasing attention from across fields, including cognitive psychology, experimental and computational

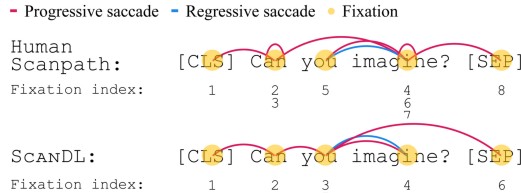

Figure 1: Human scanpath vs. SCANDL.

psycholinguistics, and computer science. As a result, computational models of eye movements in reading experienced an upsurge over the past two decades. The earlier models are explicit computational cognitive models designed for fundamental research with the aim of shedding light on i) the mechanisms underlying human language comprehension at different linguistic levels and ii) the broader question of how the human language processing system interacts with domain-general cognitive mechanisms and capacities, such as working memory or visual attention (Reichle et al., 2003; Engbert et al., 2005; Engelmann et al., 2013). More recently, traditional and neural machine learning (ML) approaches have been adopted for the prediction of human eye movements (Nilsson and Nivre, 2009, 2011; Hahn and Keller, 2016; Wang et al., 2019; Deng et al., 2023b) which, in contrast to cognitive models, do not implement any psychological or linguistic theory of eye movement control in reading. ML models, in turn, exhibit the flexibility to learn from and adapt to any kind of reading pattern on any kind of text. Within the field of ML, researchers have not only begun to *create* synthetic scanpaths but also, especially within NLP, to *leverage* them for different use cases: the interpretability of language models (LMs) (Sood et al., 2020a; Hollenstein et al., 2021, 2022; Merkx and Frank, 2021), enhancing the performance of LMs on downstream tasks (Barrett et al., 2018; Hollenstein and Zhang, 2019; Sood et al., 2020b; Deng et al., 2023a), and pre-training models for all kinds of inference tasks concerning reader- and text-specific proper-

ties (e.g., assessing reading comprehension skills or L2 proficiency, detecting dyslexia, or judging text readability) (Berzak et al., 2018; Raatikainen et al., 2021; Reich et al., 2022; Haller et al., 2022b; Hahn and Keller, 2023). For these NLP use cases, the opportunity to generate large amounts of synthetic scanpaths is crucial for two reasons: first, real human eye movement data is scarce, and its collection is resource intensive. Second, relying on real human scanpaths entails the problem of not being able to generalize beyond the respective dataset, as gaze recordings are typically not available at inference time. Synthetic scanpaths resolve both issues. However, generating synthetic scanpaths is not a trivial task, as it is a sequence-to-sequence problem that requires the alignment of two different input sequences: the word sequence (order of words in the sentence), and the scanpath (chronological order of fixations on the sentence).

In this paper, we present a novel discrete sequence-to-sequence diffusion model for the generation of synthetic human scanpaths on a given stimulus text: SCANDL, **Scan**path **D**iffusion conditioned on **L**anguage input (see Figure 1). SCANDL leverages pre-trained word representations for the text to guide the model's predictions of the location and the order of the fixations. Moreover, it aligns the different sequences and modalities (text and eye gaze) by jointly embedding them in the same continuous space, thereby capturing dependencies and interactions between the two input sequences.

The contributions of this work are manifold: We (i) develop SCANDL, the first diffusion model for simulating scanpaths in reading, which outperforms all previous state-of-the-art methods; we (ii) demonstrate SCANDL's ability to exhibit human-like reading behavior, by means of a Bayesian psycholinguistic analysis; and we (iii) conduct an extensive ablation study to investigate the model's different components, evaluate its predictive capabilities with respect to scanpath characteristics, and provide a qualitative analysis of the model's decoding process.

## 2 Related Work

**Models of eye movements in reading.** Two computational cognitive models of eye movement control in reading have been dominant in the field during the past two decades: the E-Z reader model (Reichle et al., 2003) and the SWIFT model (Engbert et al., 2005). Both models predict fixation location

and duration on a textual stimulus guided by linguistic variables such as lexical frequency and predictability. While these explicit cognitive models implement theories of reading and are designed to explain empirically observed psycholinguistic phenomena, a second line of research adopts a purely data-driven approach aiming solely at the accurate prediction of eye movement patterns. Nilsson and Nivre (2009) trained a logistic regression on manually engineered features extracted from a reader's eye movements and, in an extension, also the stimulus text to predict the next fixation (Nilsson and Nivre, 2011). More recent research draws inspiration from NLP sequence labeling tasks. For instance, Hahn and Keller (2016, 2023) proposed an unsupervised sequence-to-sequence architecture, adopting a labeling network to determine whether the next word is fixated. Wang et al. (2019) proposed a combination of CNNs, LSTMs, and a CRF to predict the fixation probability of each word in a sentence. A crucial limitation of these models is their simplifying the dual-input sequence into a single-sequence problem, not accounting for the chronological order in which the words are fixated and thus unable to predict important aspects of eye movement behavior, such as regressions and re-fixations. To overcome this limitation, Deng et al. (2023b) proposed *Eyettention*, a dual-sequence encoder-encoder architecture, consisting of two LSTM encoders that combine the word sequence and the fixation sequence by means of a cross-attention mechanism; their model predicts next fixations in an auto-regressive (AR) manner.

**Diffusion models for discrete input.** Approaches to apply diffusion processes to discrete input (mainly text) comprise discrete and continuous diffusion. Reid et al. (2022) proposed an edit-based diffusion model for machine translation and summarization whose corruption process happens in discrete space. Li et al. (2022) and Gong et al. (2023) both proposed continuous diffusion models for conditional text generation; whereas the former adopted classifiers to impose constraints on the generated sentences, the latter conditioned on the entire source sentence. All of these approaches consist of uni-modal input being mapped again to the same modality.

## 3 Problem Setting

Consider a scanpath $\mathbf{f_w^r} = \langle f_1, \ldots, f_N \rangle$, which represents a sequence of $N$ fixations generated by

| Text | [CLS] | Germans | like | fiscal | p | ##rudence | . | [SEP] | [CLS] | Germans | fiscal | prudence. | prudence. | [SEP] |
|---|---|---|---|---|---|---|---|---|---|---|---|---|---|---|
| $\mathbf{x}$ | $w_{CLS}$ | $w_1$ | $w_2$ | $w_3$ | $w_4$ | $w_5$ | $w_6$ | $w_{SEP}$ | $f_{CLS}$ | $f_1$ | $f_2$ | $f_3$ | $f_4$ | $f_{SEP}$ |
| $\mathbf{x}_{idx}$ | 0 | 1 | 2 | 3 | 4 | 4 | 4 | 5 | 0 | 1 | 3 | 4 | 4 | 5 |
| $\mathbf{x}_{bert}$ | 101 | 6494 | 1176 | 12087 | 185 | 18424 | 119 | 102 | [PAD] | [PAD] | [PAD] | [PAD] | [PAD] | [PAD] |
| $\mathbf{x}_{pos}$ | 0 | 1 | 2 | 3 | 4 | 5 | 6 | 7 | 0 | 1 | 2 | 3 | 4 | 5 |

The first eight columns are spanned by $\mathbf{x^w}$ and the last six by $\mathbf{x^f}$.

Figure 2: Discrete input representation of the concatenation $\mathbf{x}$ of sentence $\mathbf{x^w}$ and scanpath $\mathbf{x^f}$. Each element of the sequence $\mathbf{x}$ is represented by a triple of word index $\mathbf{x}_{idx}$, BERT input ID $\mathbf{x}_{bert}$, and position index $\mathbf{x}_{pos}$.

reader $r$ while reading sentence $\mathbf{w}$. Here, $f_j$ denotes the location of the $j^{th}$ fixation represented by the linear position of the fixated word within the sentence $\mathbf{w}$ (word index). The goal is to find a model that predicts a scanpath $\mathbf{f}$ given sentence $\mathbf{w}$. We evaluate the model by computing the mean Normalized Levenshtein Distance (NLD) (Levenshtein, 1965) between the predicted and the ground truth human scanpaths. Note that several readers $r$ can read the same sentence $\mathbf{w}$. In the following, we will denote the scanpath by $\mathbf{f}$ instead of $\mathbf{f_w^r}$ if the reader or sentence is unambiguous or the (predicted) scanpath is not dependent on the reader.

# 4 SCANDL

Inspired by continuous diffusion models for text generation (Gong et al., 2023; Li et al., 2022), we propose SCANDL, a diffusion model that synthesizes scanpaths conditioned on a stimulus sentence.

## 4.1 Discrete Input Representation

SCANDL uses a discrete input representation for both the stimulus sentence and the scanpath. First, we subword-tokenize the stimulus sentence $\mathbf{w} = \langle w_1, \ldots, w_M \rangle$ using the pre-trained BERT Word-Piece tokenizer (Devlin et al., 2019; Song et al., 2021). We prepend special CLS and append special SEP tokens to both the sentence and the scanpath in order to obtain the stimulus sequence $\mathbf{x^w} = \langle w_{CLS}, w_1, \ldots, w_M, w_{SEP} \rangle$ and a corresponding fixation sequence $\mathbf{x^f} = \langle f_{CLS}, f_1, \ldots, f_N, f_{SEP} \rangle$. We introduce these special tokens in order to separate the two sequences and to align their beginning and ending. The two sequences are concatenated along the sequence dimension into $\mathbf{x} = \mathbf{x^w} \oplus \mathbf{x^f}$. An example of the discrete input $\mathbf{x}$ is depicted in Figure 2 (blue row). We utilize three features to provide a discrete representation for every element in the sequence $\mathbf{x}$. The word indices $\mathbf{x}_{idx}$ align fixations in $\mathbf{x^f}$ with words in $\mathbf{x^w}$, and align subwords in $\mathbf{x^w}$ originating from the same word (yellow row in Figure 2). Second, the BERT input IDs $\mathbf{x}_{bert}$,

derived from the BERT tokenizer (Devlin et al., 2019), refer to the tokenized subwords of the stimulus sentence for $\mathbf{x^w}$, while consisting merely of PAD tokens for the scanpath $\mathbf{x^f}$, as no mapping between fixations and subwords is available (orange row in Figure 2). Finally, position indices $\mathbf{x}_{pos}$ capture the order of words within the sentence and the order of fixations within the scanpath, respectively (red row in Figure 2).

## 4.2 Diffusion Model

A diffusion model (Sohl-Dickstein et al., 2015) is a latent variable model consisting of a forward and a reverse process. In the forward process, we sample $\mathbf{z}_0$ from a real-world data distribution and gradually corrupt the data sample into standard Gaussian noise $\mathbf{z}_{\tilde{T}} \sim \mathcal{N}(0, \mathbb{I})$, where $\tilde{T}$ is the maximal number of diffusion steps. The latents $\mathbf{z}_1, \ldots, \mathbf{z}_{\tilde{T}}$ are modeled as a first-order Markov chain, where the Gaussian corruption of each intermittent noising step $t \in [1, \ldots, \tilde{T}]$ is given by $\mathbf{z}_t \sim q(\mathbf{z}_t | \mathbf{z}_{t-1}) = \mathcal{N}(\sqrt{1 - \beta_t}\mathbf{z}_{t-1}, \beta_t \mathbb{I})$, where $\beta_t \in (0, 1)$ is a hyperparameter dependent on $t$. The reverse distribution $p(\mathbf{z}_{t-1} | \mathbf{z}_t)$ gradually removes noise to reconstruct the original data sample $\mathbf{z}_0$ and is approximated by $p_{\boldsymbol{\theta}}$.

## 4.3 Forward and Backward Processes

In the following, we describe SCANDL's projection of the discrete input into continuous space, its forward noising process $q$ and reverse denoising process $p_{\boldsymbol{\theta}}$ (all depicted in Figure 3), as well as architectural details and diffusion-related methods.

### 4.3.1 Forward Process: Embedding of Discrete Input in Continuous Space

Following Gong et al. (2023), our forward process deploys an embedding function $\text{EMB}(\cdot) : \mathbb{N}^{M+N+4} \rightarrow \mathbb{R}^{(M+N+4) \times d}$ that maps from the discrete input representation into continuous space, where $N$ and $M$ denote the number of fixations and words, respectively, and $d$ is the size of the hidden dimension. This embedding learns

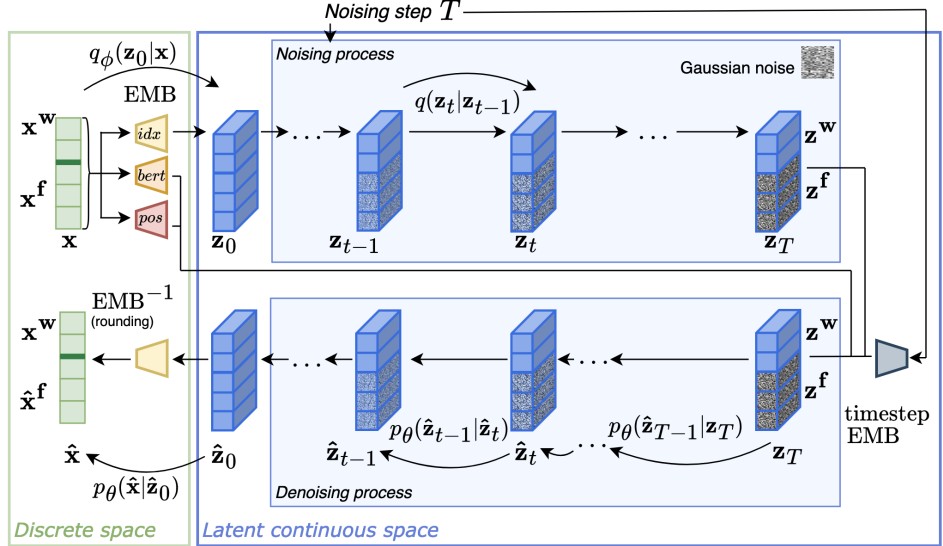

Figure 3: The embedding layer, the forward process (noising) and the reverse process (denoising) of SCANDL.

a joint representation of the subword-tokenized sentence $\mathbf{x}^{\mathbf{w}}$ and the fixation sequence $\mathbf{x}^{\mathbf{f}}$. More precisely, the embedding function $\text{EMB}(\mathbf{x}) := \text{EMB}_{idx}(\mathbf{x}_{idx}) + \text{EMB}_{bert}(\mathbf{x}_{bert}) + \text{EMB}_{pos}(\mathbf{x}_{pos})$ is the sum of three independent embedding layers (see Figure 7 in Appendix C). While the word index embedding $\text{EMB}_{idx}$ and the position embedding $\text{EMB}_{pos}$ are learned during training, the pre-trained BERT model embedding $\text{EMB}_{bert}$ is frozen. It maps the input IDs to pre-trained BERT embeddings (Devlin et al., 2019) and adds semantic meaning to the sentence. Only the word index embedding $\text{EMB}_{idx}$ is corrupted by noise during the forward process; this embedding is what the model has to learn. The other two embeddings remain unnoised.

Embedding the discrete input $\mathbf{x}$ into continuous space using our embedding function $\text{EMB}(\cdot)$ allows for a new transition $\mathbf{z}_0 \sim q_\phi(\mathbf{z}_0|\mathbf{x}) = \mathcal{N}(\text{EMB}(\mathbf{x}), \beta_0 \mathbb{I})$ to extend the original forward Markov chain, where $\mathbf{z}_0$ is the first latent variable at noising step $t = 0$ and $q_\phi$ is the parametrized embedding step of the forward process. Note that this initial latent variable $\mathbf{z}_0$ is not only a projection of the discrete data $\mathbf{x}$ into continuous space, i.e. $\text{EMB}(\mathbf{x})$, but is actually sampled from a normal distribution that is centered around $\text{EMB}(\mathbf{x})$.

### 4.3.2 Forward Process: Partial Noising

Let $\mathbf{z}_0 := \mathbf{z}_0^{\mathbf{w}} \oplus \mathbf{z}_0^{\mathbf{f}}$ be an initial latent variable, where $\mathbf{z}_0^{\mathbf{w}}$ refers to the sentence subsequence and $\mathbf{z}_0^{\mathbf{f}}$ to the fixation subsequence. Each subsequent latent $\mathbf{z}_t \in [\mathbf{z}_1, \ldots, \mathbf{z}_T]$ is given by $\mathbf{z}_t := \mathbf{z}_t^{\mathbf{w}} \oplus \mathbf{z}_t^{\mathbf{f}}$, where $\mathbf{z}_t^{\mathbf{w}}$ remains unchanged, i.e., $\mathbf{z}_t^{\mathbf{w}} = \mathbf{z}_0^{\mathbf{w}}$,

and $\mathbf{z}_t^{\mathbf{f}}$ is noised, i.e., $\mathbf{z}_t^{\mathbf{f}} \sim q(\mathbf{z}_t^{\mathbf{f}} \mid \mathbf{z}_{t-1}^{\mathbf{f}}) = \mathcal{N}(\sqrt{1-\beta_t}\mathbf{z}_{t-1}^{\mathbf{f}}, \beta_t\mathbb{I})$, with $1 \leq t \leq T$, where $T$ is the amount of noise, and $1 \leq T \leq \tilde{T}$, where $\tilde{T}$ is the maximal number of diffusion steps.[1] This *partial noising* (Gong et al., 2023) is crucial, as the sentence on which a scanpath is conditioned must remain uncorrupted.

### 4.3.3 Backward Process: Denoising

During the denoising process, a parametrized model $f_\theta$ learns to step-wise reconstruct $\mathbf{z}_0$ by denoising $\mathbf{z}_T$. Due to the first-order Markov property, the joint probability of all latents can be factorized as $p_\theta(\mathbf{z}_{0:T}) = p(\mathbf{z}_T) \prod_{t=1}^{T} p_\theta(\mathbf{z}_{t-1}|\mathbf{z}_t)$. This denoising process is modeled as $p_\theta(\mathbf{z}_{t-1}|\mathbf{z}_t) \sim \mathcal{N}(\mu_\theta(\mathbf{z}_t, t), \Sigma_\theta(\mathbf{z}_t, t))$, where $\mu_\theta(\cdot)$ and $\Sigma_\theta(\cdot)$ are the model's $f_\theta$ predicted mean and variance of the true posterior distribution $q(\mathbf{z}_{t-1}|\mathbf{z}_t)$. The optimization criterion of the diffusion model is to maximize the marginal log-likelihood of the data $\log p(\mathbf{z}_0)$. Since directly computing and maximizing $\log p(\mathbf{z}_0)$ would require access to the true posterior distribution $q(\mathbf{z}_{t-1}|\mathbf{z}_t)$, we maximize the variational lower bound (VLB) of $\log p(\mathbf{z}_0)$ as a proxy objective, defined in Equation 1. However, as SCANDL involves mapping the discrete input into continuous space and back, the training objective becomes the minimization of $\mathcal{L}_{\text{SCANDL}}$, a joint loss comprising three components, inspired by Gong et al. (2023). $\mathcal{L}_{\text{SCANDL}}$ is defined in Equation 2, where $f_\theta$ is the parametrized neural network trained to reconstruct $\mathbf{z}_0$ from $\mathbf{z}_T$ (see Section 4.5).

---

[1]Note that $T$ can be 0 (see Section 4.5); if $T = 0$, the model learns to reconstruct $\mathbf{x}$ from $\mathbf{z}_0 \sim \mathcal{N}(\text{EMB}(\mathbf{x}), \beta_0\mathbb{I})$.

$$\text{VLB} := \mathbb{E}_{q(\mathbf{z}_{1:T}|\mathbf{z}_0)} \left[ \log \frac{p(\mathbf{z}_T) \, p_{\boldsymbol{\theta}}(\mathbf{z}_0 \mid \mathbf{z}_1)}{q(\mathbf{z}_T \mid \mathbf{z}_0)} + \sum_{t=2}^{T} \log \frac{p_{\boldsymbol{\theta}}(\mathbf{z}_{t-1} \mid \mathbf{z}_t)}{q(\mathbf{z}_{t-1} \mid \mathbf{z}_t, \mathbf{z}_0)} \right] \tag{1}$$

$$\operatorname*{argmin}_{\boldsymbol{\theta}} \mathcal{L}_{\text{SCANDL}} = \operatorname*{argmin}_{\boldsymbol{\theta}} \left[ \underbrace{\sum_{t=2}^{T} \| f_{\boldsymbol{\theta}}(\mathbf{z}_t, t) - \mathbf{z}_0 \|_2^2}_{\mathcal{L}_{\text{VLB}}} + \underbrace{\| \text{EMB}(\mathbf{x}) - f_{\boldsymbol{\theta}}(\mathbf{z}_1, 1) \|_2^2}_{\mathcal{L}_{\text{EMB}}} \underbrace{- \log p_{\boldsymbol{\theta}}(\mathbf{x}|\mathbf{z}_0)}_{\mathcal{L}_{\text{round}}} \right] \tag{2}$$

The first component $\mathcal{L}_{\text{VLB}}$ is derived from the VLB (see Appendix G for a detailed derivation), and aims at minimizing the difference between ground-truth $\mathbf{z}_0$ and the model prediction $f_{\boldsymbol{\theta}}(\mathbf{z}_t, t)$. The second component $\mathcal{L}_{\text{EMB}}$ measures the difference between the model prediction $f_{\boldsymbol{\theta}}(\mathbf{z}_t, t)$ and the embedded input.[2] The last component $\mathcal{L}_{\text{round}}$ corresponds to the reverse embedding, or rounding operation, which pipes the continuous model prediction through a reverse embedding layer to obtain the discrete representation.

### 4.4 Inference

At inference time, the model $f_{\boldsymbol{\theta}}$ needs to construct a scanpath on a specific sentence $\mathbf{w}$. Specifically, to synthesize the scanpath and condition it on the embedded sentence $\text{EMB}(\mathbf{x}^{\mathbf{w}})$, we replace the word index embedding of the scanpath $\text{EMB}_{idx}(\mathbf{x^f}_{idx})$ with Gaussian noise, initializing it as $\tilde{\mathbf{x}}^{\mathbf{f}}_{idx} \sim \mathcal{N}(0, \mathbb{I})$. We then concatenate the new embedding $\widetilde{\text{EMB}}(\mathbf{x^f}) = \tilde{\mathbf{x}}^{\mathbf{f}}_{idx} + \text{EMB}_{bert}(\mathbf{x^f}_{bert}) + \text{EMB}_{pos}(\mathbf{x^f}_{pos})$ with $\text{EMB}(\mathbf{x^w})$ to obtain the model input $\mathbf{z}_{\tilde{T}}$. The model $f_{\boldsymbol{\theta}}$ then iteratively denoises $\mathbf{z}_{\tilde{T}}$ into $\mathbf{z}_0$. At each denoising step $t$, an anchoring function is applied to $\mathbf{z}_t$ that serves two different purposes. First, it performs rounding on $\mathbf{z}_t$ (Li et al., 2022), which entails mapping it into discrete space and then projecting it back into continuous space so as to enforce intermediate steps to commit to a specific discrete representation. Second, it replaces the part in $\mathbf{z}_{t-1}$ that corresponds to the condition $\mathbf{x^w}$ with the original $\text{EMB}(\mathbf{x^w})$ (Gong et al., 2023) to prevent the condition from being corrupted by being recovered by the model $f_{\boldsymbol{\theta}}$. After denoising $\mathbf{z}_{\tilde{T}}$ into $\mathbf{z}_0$, $\mathbf{z}_0$ is piped through the inverse embedding layer to obtain the predicted scanpath $\hat{\mathbf{x}}^{\mathbf{f}}$.

### 4.5 Model and Diffusion Parameters

Our parametrized model $f_{\boldsymbol{\theta}}$ consists of an encoder-only Transformer (Vaswani et al., 2017) comprising

12 blocks, with 8 attention heads and hidden dimension $d = 256$. An extra linear layer projects the pre-trained BERT embedding $\text{EMB}_{bert}$ to the hidden dimension $d$. The maximum sequence length is 128, and the number of diffusion steps is $\tilde{T} = 2000$. We use a *sqrt* noise schedule (Li et al., 2022) to sample $\beta_t = 1 - \sqrt{\frac{t}{\tilde{T}+s}}$, where $s = 0.0001$ is a small constant corresponding to the initial noise level. To sample the noising step $T \in \left[0, 1, \ldots, \tilde{T}\right]$, we employ importance sampling as defined by Nichol and Dhariwal (2021).

## 5 Experiments

To evaluate the performance of SCANDL against both cognitive and neural scanpath generation methods, we perform a within- and an across-dataset evaluation. All models were implemented in PyTorch (Paszke et al., 2019), and trained for 80,000 steps on four NVIDIA GeForce RTX 3090 GPUs. For more details on the training, see Appendix A. Our code is publicly available at https://github.com/DiLi-Lab/ScanDL.

### 5.1 Datasets

We use two eye-tracking-while-reading corpora to train and/or evaluate our model. The *Corpus of Eye Movements in L1 and L2 English Reading* (CELER, Berzak et al., 2022) is an English sentence corpus including data from native (L1) and non-native (L2) English speakers, of which we only use the L1 data (CELER L1). The *Zurich Cognitive Language Processing Corpus* (ZuCo, Hollenstein et al., 2018) is an English sentence corpus comprising both "task-specific" and "natural" reading, of which we only include the natural reading (ZuCo NR). Descriptive statistics for the two corpora including the distribution of reading measures and participant demographics can be found in Section B of the Appendix.

---

[2]Recall that $\text{EMB}(\mathbf{x}) \neq \mathbf{z}_0$, as $\mathbf{z}_0 \sim \mathcal{N}(\text{EMB}(\mathbf{x}), \beta_0 \mathbb{I})$.

## 5.2 Reference Methods

We compare SCANDL with other state-of-the-art approaches to generate synthetic scanpaths including two well-established cognitive models, the E-Z reader model (Reichle et al., 2003) and the SWIFT model (Engbert et al., 2005), and one machine-learning-based model, Eyettention (Deng et al., 2023b). Moreover, we include a human baseline, henceforth referred to as *Human*, that measures the inter-reader scanpath similarity for the same sentence. Finally, we compare the model with two trivial baselines. One is the Uniform model, which simply predicts fixations iid over the sentence, and the other one, referred to as Train-label-dist model, samples the saccade range from the training label distribution (Deng et al., 2023b).

## 5.3 Evaluation Metric

To assess the model performance, we compute the Normalized Levenshtein Distance (NLD) between the predicted and the human scanpaths. The Levenshtein Distance (LD, Levenshtein, 1965) is a similarity-based metric quantifying the minimal number of additions, deletions and substitutions required to transform a word-index sequence $S$ of a true scanpath into a word-index sequence $\hat{S}$ of the model-predicted scanpath. Formally, the NLD is defined as $\text{NLD}(S, \hat{S}) = \text{LD}/\max(|S|, |\hat{S}|)$.

## 5.4 Hyperparameter Tuning

To find the best model-specific and training-specific hyperparameters of SCANDL, we perform triple cross-validation on the *New Reader/New Sentence* setting (for the search space, see Appendix A).

## 5.5 Within-Dataset Evaluation

For the within-dataset evaluation, we perform 5-fold cross-validation on CELER L1 (Berzak et al., 2022) and evaluate the model in 3 different settings. The results for all settings are provided in Table 1.

*New Sentence* **setting.** We investigate the model's ability to generalize to novel sentences read by known readers; i.e., the sentences in the test set have not been seen during training, but the readers appear both in the training and test set.

   **Results.** SCANDL not only outperforms the previous state-of-the-art Eyettention (Deng et al., 2023b) as well as all other reference methods by a significant margin, but even exceeds the similarity that is reached by the Human baseline.

*New Reader* **setting.** We test the model's ability to generalize to novel readers; the test set consists of scanpaths from readers that have not been seen during training, although the sentences appear both in training and test set.

   **Results.** Again, our model both improves over the previous state-of-the-art, the cognitive models, as well as over the Human baseline. Even more, the model achieves a greater similarity on novel readers as compared to novel sentences in the previous setting.

*New Reader/New Sentence* **setting.** The test set comprises only sentences and readers that the model has not seen during training to assess the model's ability to simultaneously generalize to novel sentences and novel readers. Of the within-dataset settings, this setting exhibits the most out-of-distribution qualities.

   **Results.** Again, SCANDL significantly outperforms the previous state-of-the-art. Again, in contrast to previous approaches and in line with the other settings, SCANDL attains higher similarity as measured in NLD than the Human baseline.

## 5.6 Across-Dataset Evaluation

To evaluate the generalization capabilities of our model, we train it on CELER L1 (Berzak et al., 2022), while testing it across-dataset on ZuCo NR (Hollenstein et al., 2018). Although the model has to generalize to unseen readers and sentences in the *New Reader/New Sentence* setting, the hardware setup and the presentation style including stimulus layout of the test data are the same, and the readers stem from the same population. In the *Across-Dataset* evaluation, therefore, we not only examine the model's ability to generalize to novel readers and sentences, but also to unfamiliar hardware and presentation style.

   **Results.** The results for this setting can also be found in Table 1. SCANDL outperforms all reference models, and achieves similarity with a true scanpath on par with the Human baseline.

## 5.7 Ablation Study

In this section, we investigate the effect of omitting central parts of the model: SCANDL without the position embedding $\text{EMB}_{pos}$ and the pre-trained BERT embedding $\text{EMB}_{bert}$, and SCANDL without the sentence condition (unconditional scanpath generation). Additionally, we also consider the two previously prevalent noise schedules, the linear

| Model | New Sentence | New Reader | New Reader/New Sentence | Across-Dataset |
|---|---|---|---|---|
| Uniform | 0.779 ± 0.002† | 0.781 ± 0.003† | 0.782 ± 0.005† | 0.802 |
| Train-label-dist | 0.672 ± 0.003† | 0.672 ± 0.004† | 0.674 ± 0.005† | 0.723 |
| E-Z Reader | 0.619 ± 0.005† | 0.620 ± 0.006† | 0.622 ± 0.006† | 0.667 |
| SWIFT | 0.607 ± 0.004† | 0.608 ± 0.006† | 0.607 ± 0.006† | 0.703 |
| Eyettention | 0.580 ± 0.002† | 0.580 ± 0.004† | 0.578 ± 0.006† | 0.697 |
| **SCANDL** | **0.516 ± 0.006** | **0.509 ± 0.014** | **0.515 ± 0.014** | **0.647** |
| *Human* | *0.538 ± 0.006* | *0.536 ± 0.004* | *0.538 ± 0.006* | *0.646 ± 0.002* |

Table 1: We report NLD ± standard error for all settings. The dagger † indicates models significantly worse than the best model. In the *New Sentence*, *New Reader*, and *New Reader/New Sentence* settings, models are evaluated using five-fold cross-validation. In the *Across-Dataset* setting, the model is trained on CELER L1 and tested on ZuCo NR.

| Ablation case | NLD ↓ |
|---|---|
| **SCANDL** (original) | **0.515 ± 0.014** |
| Cosine | **0.514 ± 0.018** |
| Linear | 0.519 ± 0.020 |
| W/o condition | 0.667 ± 0.015 |
| W/o $\text{EMB}_{bert}$ and $\text{EMB}_{pos}$ | 0.968 ± 0.002 |

Table 2: *Ablation study.* We report NLD ± standard error for 5-fold cross-validation in the *New Reader/New Sentence* setting.

(Ho et al., 2020) and the cosine (Nichol and Dhariwal, 2021) noise schedules (for their definition, see Appendix C.2). All ablation cases are conducted in the *New Reader/New Sentence* setting.

**Results.** As shown in Table 2, omitting all embeddings except for the word index embedding results in a significant performance drop, as well as training SCANDL on unconditional scanpath generation. Moreover, changing the *sqrt* to a linear noise schedule does not enhance performance, while there is a slight increase in performance for the cosine noise schedule. However, this performance difference between the *sqrt* and the cosine schedule is statistically not significant.[3]

## 6 Investigation of Model Behavior

### 6.1 Psycholinguistic Analysis

We further assess SCANDL's ability to exhibit human-like gaze behavior by investigating psycholinguistic phenomena observed in humans. We compare the effect estimates of three well-established psycholinguistic predictors — *word length, surprisal* and *lexical frequency effects* — on a range of commonly analyzed reading measures — first-pass regression rate (FPR), skipping rate (SR), first-pass fixation counts (FFC) and total fixation counts (TFC) — between human scanpaths on the one hand and synthetic scanpaths generated by SCANDL and our reference methods on the other

hand.[4] Effect sizes are estimated using Bayesian generalized linear-mixed models with reading measures as target variables and psycholinguistic features as predictors; logistic models for FPR and SR, Poisson models for FFC and TFC. For the human data, we fit random intercepts.[5] We compute posterior distributions over all effect sizes using brms (Bürkner, 2017), running 4 chains with 4000 iterations including 1000 warm-up iterations.

**Results.** We present the posterior distributions of the effect estimates obtained for the four reading measures in the *New Reader/New Sentence* setting in Figure 4 and in Table 7 of the Appendix. We observe that across all reading measures and psycholinguistic predictor variables, SCANDL exhibits effects that are most consistent with the human data. On the one hand, the qualitative pattern, i.e., the sign of the estimates, for SCANDL-scanpaths is identical to the pattern observed in the human data, outperforming not only previous state-of-the-art (SOTA) models such as Eyettention, but even the two arguably most important cognitive models of eye movements in reading, E-Z reader and SWIFT. On the other hand, also the quantitative pattern, i.e., the estimated *size* of each of the effects, exhibited by ScanDL are most similar to the ones observed in humans.

### 6.2 Emulation of Reading Pattern Variability

To assess whether SCANDL is able to emulate the different reading patterns and their variability typical for human data, we compare the true and the predicted scanpaths in the *New Reader* setting of both datasets with respect to the mean and standard deviation of a range of commonly analyzed reading measures: regression rate, normalized fixation

---

[3]$p = 0.68$ in a paired *t*-test.

[4]Full results plots, including the (psycholinguistically irrelevant) Uniform and Train-label-dist baselines, can be found in Fig. 8 of the Appendix.

[5]For more details on the reading measures, computation of the predictors and model specification, see Appendix D.

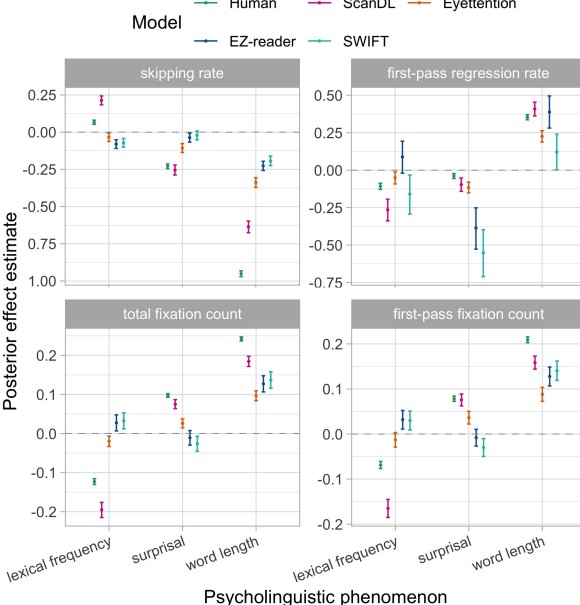

Figure 4: Comparison of posterior effect estimates for psycholinguistic phenomena on reading measures between original and predicted scanpaths. Lines represent a 95% credible interval, means are denoted by dots.

count, progressive and regressive saccade length, skipping rate, and first-pass count (see Table 8 in the Appendix for a detailed definition).

**Results.** As depicted in Figure 5 and Table 9 in the Appendix, the scanpaths generated by SCANDL are similar in diversity to the true scanpaths for both datasets. Not only is SCANDL's mean value of each reading measure close to the true data, but, crucially, the model also reproduces the variability in the scanpaths: for instance, in both datasets, the variability is big in both the true and the predicted scanpaths for regressive saccades, and is small in both true and predicted scanpaths with regards to progressive saccade lengths and first-pass counts.

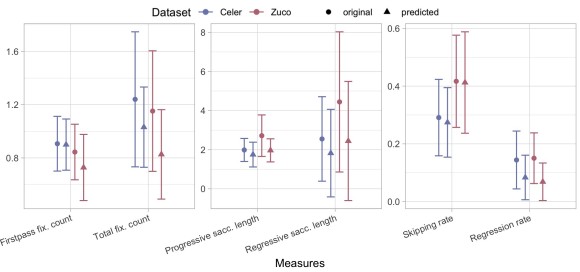

Figure 5: Reading measures of true and predicted scanpaths by SCANDL of the CELER and ZuCo datasets.

We also inspect the model's ability to approximate reader-specific patterns. To this end, we average the above analyzed reading measures over all scanpaths of a reader, and compute the correlation

with this reader's mean NLD. The analysis suggests that SCANDL more easily predicts patterns of readers with shorter scanpaths and a low proportion of regressions. Exact numbers are displayed in Table 10 in the Appendix.

### 6.3 Qualitative Differences Between Models

To investigate whether SCANDL and the baseline models exhibit the same qualitative pattern in their predictive performance, that is, whether they generate good (or poor) predictions on the same sentences, we compute the correlations between the mean sentence NLDs of SCANDL and each of the reference models. The correlations are all significant, ranging from 0.36 (SWIFT) to 0.41 (Eyettention), though neither very strong nor wide-spread (see Table 11 in the Appendix). For a more detailed inspection of each model's ability to predict certain reading patterns (scanpath properties), we compute the correlations between the above-introduced reading measures (see Section 6.2) of a true scanpath and the NLD of its predicted counterpart for all models, in the *New Reader/New Sentence* setting.

**Results.** All models exhibit a significant positive correlation between the NLD and both the regression rate and the normalized fixation count (see Table 12 in the Appendix): long scanpaths with many regressions and a high relative number of fixations result in predictions with a higher NLD (i.e., are more difficult). Further, models of the same type, that is the two ML-based models SCANDL and Eyettention on the one hand, and the two cognitive models E-Z reader and SWIFT on the other hand, exhibit very similar patterns. Overall, the cognitive models have much stronger correlations than the ML-based models. In particular, E-Z reader and SWIFT exhibit a large positive correlation for first-pass counts, meaning that they struggle with scanpaths with a high number of first-pass fixations on the words, while both SCANDL and Eyettention have correlations close to zero, indicating that they cope equally well for scanpaths with high or low first-pass counts. In sum, while the cognitive models appear to be overfitting to specific properties of a scanpath, SCANDL and Eyettention seem to generalize better across patterns.

### 6.4 Investigation of the Decoding Progress

We further examine the denoising process of SCANDL to determine at which step the Gaussian noise is shaped into the embeddings representing the word IDs of the scanpath. Figure 6 depicts three

t-SNE plots (Hinton and Roweis, 2002) at different steps of the decoding process. Only during the last 200 denoising steps do we see an alignment between the words in the sentence and the predicted fixations on words in the scanpath, and a clear separation between these predicted fixations and the predicted PAD tokens. At the very last denoising step, all PAD tokens are mapped to a small number of spatial representations.

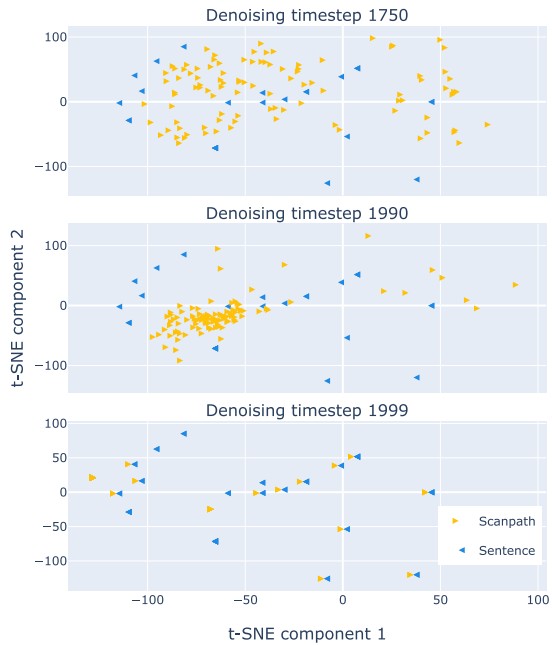

Figure 6: t-SNE plots of the continuous model output $\hat{\mathbf{z}}_t$ at different steps of the 2000-step denoising process. Step 1999 refers to the last iteration (all noise removed).

## 7 Discussion

The experimental results show that our model establishes a new state-of-the-art in generating human-like synthetic eye movement patterns across all investigated evaluation scenarios, including within- and across-dataset settings. Indeed, the similarity of the model's synthetic scanpaths to a human scanpath even exceeds the similarity between two scanpaths generated by two different human readers on the same stimulus. This indicates that the model learns to mimic an average human reader, abstracting away from reader-specific idiosyncrasies.[6] Interestingly, the model's performance is better in the *New Reader* than in both the *New Sentence* and *New Reader/New Sentence* setting, which stands in stark contrast to previous research which identified the generalization to novel readers as the

---

[6]This is further supported by the NLD not changing when computed as average NLD between a SCANDL scanpath and *all* human scanpaths on the same sentence in the test set.

main challenge (Reich et al., 2022). Furthermore, SCANDL's SOTA performance in the *Across-Dataset* setting, attaining parity with the human baseline, corroborates the model's generalizability. This generalizability is further underlined by the model emulating the variability in reading patterns observed in the human data, even when being evaluated across-dataset.

The omission of the positional embedding and the pre-trained BERT embedding in the ablation study highlights their importance — the fact that discarding them yields a worse performance than omitting the sentence condition, in which case the model still receives the positional embedding, stresses the importance of sequential information, which is lost to a transformer model if not explicitly provided. Moreover, removing the sentence-condition also emphasizes the importance of the linguistic information contained in the sentence. Overall, the ablation study emphasizes the importance of modeling scanpath prediction as a dual-nature problem.

In contrast to cognitive models, SCANDL was not designed to exhibit the same psycholinguistic phenomena as human readers. However, our psycholinguistic analysis demonstrates that SCANDL nevertheless captures the key phenomena observed in human readers which psycholinguistic theories build on. Even more, in contrast to the cognitive models, it appears to overfit less to certain reading patterns. These findings not only emphasize the high quality of the generated data, but also open the possibility to use SCANDL when designing psycholinguistic experiments: the experimental stimuli can be piloted by means of simulations with SCANDL to potentially detect unexpected patterns or confounds and address them before conducting the actual experiment with human readers. Further, we can use SCANDL for human-centric NLG evaluations using synthetic scanpaths.

## 8 Conclusion

We have introduced a new state-of-the-art model for scanpath generation called SCANDL. It not only resolves the two major bottlenecks for cognitively enhanced and interpretable NLP, data scarcity and unavailability at inference time, but also promises to be valuable for psycholinguistics by producing high quality human-like scanpaths. Further, we have extended the application of diffusion models to discrete sequence-to-sequence problems.

## Limitations

Eye movement patterns in reading exhibit a high degree of individual differences between readers (Kuperman and Van Dyke, 2011; Jäger et al., 2020; Haller et al., 2022a, 2023). For a generative model of scanpaths in reading, this brings about a trade-off between group-level predictions and predictions accounting for between-reader variability. The fact that SCANDL outperforms the human baseline in terms of NLD indicates that it learns to emulate an average reader. Whereas this might be the desired behavior for a range of use case scenarios, it also means that the model is not able to concomitantly predict the idiosyncrasies of specific readers. We plan to address this limitation in future work by adding reader-specific information to the model.

Relatedly, since our model has been trained and evaluated on a natural reading task, it remains unclear as to what extent it generalizes to task-specific datasets, which arguably might provide more informative scanpaths for the corresponding NLP downstream task. As for the reader-specific extension of the model, this issue might be addressed by adding the task as an additional input condition.

On a more technical note, a major limitation of the presented model is its relatively high computational complexity in terms of run time and memory at inference time (see Section A.2 in the Appendix).

Moreover, the metric used for model evaluation, the Normalized Levensthein Distance, might not be the ideal metric for evaluating scanpaths. Other metrics that have been used to measure scanpath similarity — MultiMatch (Jarodzka et al., 2010) and ScanMatch (Cristino et al., 2010) — have been questioned in terms of their validity in a recent study (Kümmerer and Bethge, 2021); both metrics have systematically scored incorrect models higher than ground-truth models. A better candidate to use in place of the Normalized Levenshtein Distance might be the similarity score introduced by von der Malsburg et al. (2015), which has been shown to be sensitive to subtle differences between scanpaths on sentences that are generally deemed simple. However, the main advantage of this metric is that it takes into account fixation durations, which SCANDL, in its current version, is unable to predict.

This inability to concomitantly predict fixation durations together with the fixation positions is another shortcoming of our model. However, we aim to tackle this problem in future work.

Finally, we would like to emphasize that, although our model is able to capture psycholinguistic key phenomena of human sentence processing, it is *not* a cognitive model and hence does not claim in any way that its generative process simulates or resembles the mechanisms underlying eye movement control in humans.

## Ethics Statement

Working with human data requires careful ethical consideration. The eye-tracking corpora used for training and testing follow ethical standards and have been approved by the responsible ethics committee. However, in recent years it has been shown that eye movements are a behavioral biometric characteristic that can be used for user identification, potentially violating the right of privacy (Jäger et al., 2020; Lohr and Komogortsev, 2022). The presented approach of using synthetic data at deployment time considerably reduces the risk of potential privacy violation, as synthetic eye movements do not allow to draw any conclusions about the reader's identity. Moreover, the model's capability to generate high-quality human-like scanpaths reduces the need to carry out eye-tracking experiments with humans across research fields and applications beyond the use case of gaze-augmented NLP. Another possible advantage of our approach is that by leveraging synthetic data, we can overcome limitations associated with the availability and representativeness of real-world data, enabling the development of more equitable and unbiased models.

In order to train high-performing models for downstream tasks using gaze data, a substantial amount of training data is typically required. However, the availability of such data has been limited thus far. The utilization of synthetic data offers a promising solution by enabling the training of more robust models for various downstream tasks using gaze data. Nonetheless, the adoption of this approach raises important ethical considerations, as it introduces the potential for training models that can be employed across a wide range of tasks, including those that may be exploited for nefarious purposes. Consequently, there exists a risk that our model could be utilized for tasks that are intentionally performed in bad faith.

## Acknowledgements

This work was partially funded by the German Federal Ministry of Education and Research under grant 01| S20043 and the Swiss National Science Foundation under grant 212276, and is supported by COST Action MultiplEYE, CA21131, supported by COST (European Cooperation in Science and Technology).

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

# Appendix for SCANDL: A Diffusion Model for Generating Synthetic Scanpaths on Texts

## A  Parameters and Hyperparameters

### A.1  Diffusion-specific Hyperparameters

Hyperparameters concerning SCANDL or the noising process include the number of encoder blocks, the number of attention heads in each attention layer, the hidden dimension of the transformer, the noise schedule, the schedule sampling, the number of diffusion steps, and the input sequence length. We set the number of diffusion steps to be $\tilde{T} = 2,000$, the input sequence length to 128, and opt for importance sampling (Nichol and Dhariwal, 2021) as concerns the schedule sampling. The other hyperparameters are determined during hyperparameter tuning; the search space as well as the best values can be found in Table 3.

Concerning the noise schedules specifically, we list and define the ones we investigate below:
- linear noise schedule: $\beta_t = 10^{-4} * (1 - i) + 0.02 * i$, with $i = \frac{t-1}{\tilde{T}-1}$,
- *sqrt* noise schedule: $\beta_t = 1 - \sqrt{\frac{t}{\tilde{T}+s}}$, where $s$ is a constant corresponding to the starting noise level, which we set to $s = 0.0001$,
- cosine noise schedule: $\beta_t = \frac{f(t)}{f(0)}$, with $f(t) = \cos(\frac{t/\tilde{T}+0.008}{1.008} * \frac{\pi}{2})^2$,
- truncated cosine noise schedule: $\beta_t = \frac{f(t)}{f(0)}$, with $f(t) = \cos(\frac{t/\tilde{T}+0.008}{1.008} * \frac{\pi}{2})^2$,
- truncated linear noise schedule: $\beta_t = (10^{-4} + 0.01) * (1 - i) + (0.02 + 0.01) * i$, with $i = \frac{t-1}{\tilde{T}-1}$.

We uniformly sample 60 model configurations that are evaluated in a triple cross-validation manner in the *New Reader/New Sentence* setting, and we employ the normalized Levenshtein Distance (Levenshtein, 1965) as selection criterion.

| Parameter | Search space | Best value |
|---|---|---|
| Number of encoder blocks | {2, 4, 8, 12, 16} | 12 |
| Number of attention heads | {2, 4, 8, 12, 16} | 8 |
| Hidden dimension $d$ | {128, 256, 512, 768} | 256 |
| Noise schedule | {linear, *sqrt*, cosine, truncated cosine, truncated linear} | *sqrt* |

Table 3: Parameters and search space used for hyperparameter tuning.

### A.2  Training-specific Hyperparameters

We train SCANDL for 80,000 learning steps in a parallel manner over 4 NVIDIA GeForceRTX 3090 GPUs using the AdamW optimizer (Loshchilov and Hutter, 2019), a learning rate of 1e-4, and a batch size of 64.

### A.3  Training Time, Inference Time, Parameters

SCANDL consists of about 130 million trainable parameters[7], with an exception for the ablation case of SCANDL without $\text{EMB}_{bert}$ and $\text{EMB}_{pos}$ and the ablation case without the sentence condition, which both consist of about 22 million trainable parameters. Training the above-mentioned configuration of SCANDL for 80,000 learning steps takes about 8 hours. The inference/decoding is done on one NVIDIA GeForceRTX 3090 GPU and takes about 75 minutes for 2200 sentences, which is around 2 seconds per sentence.

## B  Datasets

As regards reader-specific demographic information in CELER (Berzak et al., 2022), there are 69 L1 English speakers in CELER, of which 38 are female and 28 are male, one identifying as "other" and two

---

[7]130,855,296 parameters

not divulging their gender. The mean age is $26.3 \pm 6.7$ years. Participants were recruited via a variety of sources, such as mailing lists, advertisements on social media, or message boards. With respect to ZuCo (Hollenstein et al., 2018), there are 12 participants, of which 5 are female and 7 male, all native English speakers between the ages of 25 and 51, with a mean age of $35 \pm 9.8$ years.

Descriptive statistics for the two corpora used in our experiments can be found in Table 4. Descriptive statistics on mean reading measures values in the L1 part of CELER Berzak et al. (2022) can be found in Table 5

| Dataset | Eye-tracker (sampling freq.) | # Unique sentences | # Words per sentence | # Readers |
|---|---|---|---|---|
| CELER L1 (Berzak et al., 2022) | EyeLink 1000 (1000 Hz) | 5456 | $11.2 \pm 3.6$ | 69 |
| ZuCo NR (Hollenstein et al., 2018) | EyeLink 1000 Plus (500 Hz) | 700 | $19.6 \pm 9.8$ | 12 |

Table 4: Descriptive statistics of the two eye-tracking corpora used for model training and evaluation, CELER (Berzak et al., 2022) and ZuCo (Hollenstein et al., 2018). The number of words per sentence is reported using the mean $\pm$ standard deviation.

| Measure | Value |
|---|---|
| Number of fixations per word | $1.3 \pm 0.1$ |
| Saccade length in chars | $8.5 \pm 0.4$ |
| Skip rate | $0.36 \pm 0.02$ |
| Regression rate | $0.24 \pm 0.01$ |

Table 5: Mean eye movement measures with 95 % confidence intervals of the L1 speakers in CELER Berzak et al. (2022).

# C   Implementation Details

## C.1   Embedding Figure

Figure 7 shows the embedding function $\text{EMB}(\mathbf{x})(\cdot): \mathbb{N}^{M+N+4} \rightarrow \mathbb{R}^{(M+N+4) \times d}$ that maps from the discrete input representation into continuous space, where $d$ is the size of the hidden dimension. This embedding learns a joint representation of the subword-tokenized sentence $\mathbf{x^w}$ and the fixation sequence $\mathbf{x^f}$. More precisely, the embedding function $\text{EMB}(\mathbf{x}) \coloneqq \text{EMB}_{idx}(\mathbf{x}_{idx}) + \text{EMB}_{bert}(\mathbf{x}_{bert}) + \text{EMB}_{pos}(\mathbf{x}_{pos})$ is the sum of three independent embedding layers.

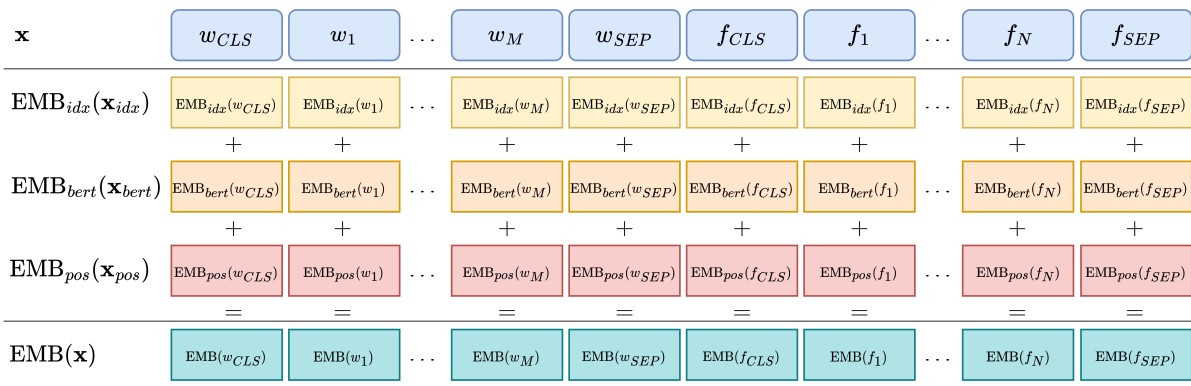

Figure 7: Embeddings of the discrete input representations.

## C.2   Ablation: Noise Schedules

For the ablation study, compare the *sqrt* noise schedule (Li et al., 2022) with the linear noise schedule (Ho et al., 2020) and the cosine noise schedule (Nichol and Dhariwal, 2021). The definitions of these noise schedules can be found in Table 6.

| Noise schedule | Definition |
|---|---|
| Sqrt | $\beta_t = 1 - \sqrt{\frac{t}{\tilde{T}+s}}$, with $s = 0.0001$ |
| Linear | $\beta_t = 10^{-4} * (1-i) + 0.02 * i$, with $i = \frac{t-1}{\tilde{T}-1}$ |
| Cosine | $\beta_t = \frac{f(t)}{f(0)}$, with $f(t) = \cos(\frac{t/\tilde{T}+0.008}{1.008} * \frac{\pi}{2})^2$ |

Table 6: Noise schedules used for SCANDL and the ablation study.

## D Psycholinguistic Analysis

### D.1 Reading Measures and Predictors

#### D.1.1 Reading Measures

The reading measures used for the psycholinguistic analysis are defined as follows:

- first-pass regression rate (FPR; binary): 1 if a regression was initiated for a given word when visiting it the first time, else 0.
- Skipping rate (SR; binary): 1 if the word was skipped when visited for the first time.
- first-pass fixation counts (FFC): Number of consecutive fixations on a word when visiting it the first time.
- total fixation counts (TFC): Total number of fixations on a given word.

#### D.1.2 Psycholinguistic Predictors

The psycholinguistic predictors were computed as follows:

- Lexical frequency: Frequencies were extracted using the python library wordfreq: https://github.com/rspeer/wordfreq
- Suprisal: For each token $w_{jk}$ in a given sentence $j$, surprisal is defined as $-\log p(w_{jk} \mid \mathbf{w}_{j<k})$. We extracted surprisal values from the GPT-2 language model gpt-xl provided by Hugging Face (Radford et al., 2019).

### D.2 Model specification

We model the reading measures $y$ using Bayesian generalized linear models – logistic regression models for the binary reading measures (skipping rate and first-pass regression rate), Poisson regression models for the count-based reading measures (total fixation counts and first-pass fixation counts). The predictors are word lenght (wl), lexical frequeny (freq) and surprisal (surp). For the human data, we include random intercepts for subjects.

$$y_{ij} = g\left(\beta_0 + \beta_{0i} + \beta_1\, wl_j + \beta_2\, freq_j + \beta_3\, surp_j\right) \tag{3}$$

where $y_{ij}$ refers to the reading measure of subject $i$ for the $j$th word in a given sentence. $\beta_0$ represents the global intercept, and $\beta_{0i}$ the random intercept for subject $i$. $g(\cdot)$ denotes the linking function: $g(z) = \ln\frac{z}{1-z}$ for the binary measures, with $y_{ij}$ following a Bernoulli distribution; and $g(z) = \log z$ for the count-based measures, with $y_{ij}$ following a Poisson distribution. For the predicted data, we fit the same model, but without by-subject random intercepts. As priors, we use the brms standard priors.

### D.3 Full Results

In addition to the figures in the main part, Fig. 8 contains shows posterior effect estimates for all baselines. Summary statistics (mean and Bayesian 95%-credible intervals) of the effect estimate posterior distributions in the *New Reader/New Sentence* setting can be found in Table 7.

| | Measure | Model | Mean | 95% CI |
|---|---|---|---|---|
| *lexical frequency* | SR | Human | 0.07 | [0.05, 0.08] |
| | | ScanDL | 0.21 | [0.18, 0.24] |
| | | Eyettention | −0.03 | [−0.06, −0.01] |
| | | E-Z reader | −0.08 | [−0.11, −0.05] |
| | | SWIFT | −0.07 | [−0.10, −0.04] |
| | | Train-label-dist | −0.04 | [−0.07, −0.01] |
| | FPR | Human | −0.11 | [−0.13, −0.09] |
| | | ScanDL | −0.26 | [−0.34, −0.19] |
| | | Eyettention | −0.05 | [−0.09, −0.01] |
| | | E-Z reader | 0.09 | [−0.02, 0.19] |
| | | SWIFT | −0.16 | [−0.29, −0.03] |
| | | Train-label-dist | −0.11 | [−0.16, −0.05] |
| | | Uniform | −0.03 | [−0.11, 0.06] |
| | TFC | Human | −0.12 | [−0.13, −0.12] |
| | | ScanDL | −0.20 | [−0.21, −0.18] |
| | | Eyettention | −0.02 | [−0.03, −0.01] |
| | | E-Z reader | 0.03 | [0.01, 0.05] |
| | | SWIFT | 0.03 | [0.01, 0.05] |
| | | Train-label-dist | 0.03 | [0.02, 0.05] |
| | | Uniform | 0.19 | [0.17, 0.22] |
| | FFC | Human | −0.07 | [−0.08, −0.06] |
| | | ScanDL | −0.17 | [−0.19, −0.15] |
| | | Eyettention | −0.01 | [−0.03, 0.00] |
| | | E-Z reader | 0.03 | [0.01, 0.05] |
| | | SWIFT | 0.03 | [0.01, 0.05] |
| | | Train-label-dist | 0.04 | [0.02, 0.06] |
| | | Uniform | 0.17 | [0.14, 0.20] |
| *surprisal* | SR | Human | −0.23 | [−0.24, −0.21] |
| | | ScanDL | −0.25 | [−0.29, −0.22] |
| | | Eyettention | −0.11 | [−0.14, −0.08] |
| | | E-Z reader | −0.04 | [−0.07, −0.01] |
| | | SWIFT | −0.02 | [−0.05, 0.01] |
| | | Train-label-dist | −0.15 | [−0.18, −0.12] |
| | | Uniform | −0.26 | [−0.30, −0.22] |
| | FPR | Human | −0.04 | [−0.06, −0.02] |
| | | ScanDL | −0.10 | [−0.14, −0.05] |
| | | Eyettention | −0.12 | [−0.15, −0.08] |
| | | EZ reader | −0.39 | [−0.53, −0.25] |
| | | SWIFT | −0.55 | [−0.71, −0.40] |
| | | Train-label-dist | −0.04 | [−0.09, 0.01] |
| | | Uniform | −0.08 | [−0.15, 0.00] |
| | TFC | Human | 0.10 | [0.09, 0.10] |
| | | ScanDL | 0.08 | [0.06, 0.09] |
| | | Eyettention | 0.03 | [0.01, 0.04] |
| | | E-Z reader | −0.01 | [−0.03, 0.01] |
| | | SWIFT | −0.03 | [−0.05, −0.01] |
| | | Train-label-dist | 0.13 | [0.11, 0.14] |
| | | Uniform | 0.36 | [0.34, 0.39] |
| | FFC | Human | 0.08 | [0.07, 0.08] |
| | | ScanDL | 0.08 | [0.06, 0.09] |
| | | Eyettention | 0.04 | [0.02, 0.05] |
| | | E-Z reader | −0.01 | [−0.03, 0.01] |
| | | SWIFT | −0.03 | [−0.05, −0.01] |
| | | Train-label-dist | 0.11 | [0.09, 0.13] |
| | | Uniform | 0.33 | [0.30, 0.36] |

| | Measure | Model | Mean | 95% CI |
|---|---|---|---|---|
| *word length* | SR | Human | −0.95 | [−0.97, −0.93] |
| | | SCANDL | −0.64 | [−0.68, −0.60] |
| | | Eyettention | −0.34 | [−0.37, −0.31] |
| | | E-Z reader | −0.23 | [−0.26, −0.20] |
| | | SWIFT | −0.19 | [−0.23, −0.16] |
| | | Train-label-dist | 0.08 | [0.05, 0.11] |
| | | Uniform | 0.13 | [0.08, 0.18] |
| | FPR | Human | 0.35 | [0.34, 0.37] |
| | | SCANDL | 0.41 | [0.36, 0.45] |
| | | Eyettention | 0.23 | [0.19, 0.26] |
| | | E-Z reader | 0.39 | [0.28, 0.50] |
| | | SWIFT | 0.12 | [0.00, 0.24] |
| | | Train-label-dist | −0.02 | [−0.07, 0.03] |
| | | Uniform | 0.10 | [0.02, 0.18] |
| | TFC | Human | 0.24 | [0.24, 0.25] |
| | | SCANDL | 0.18 | [0.17, 0.20] |
| | | Eyettention | 0.10 | [0.08, 0.11] |
| | | E-Z reader | 0.13 | [0.11, 0.15] |
| | | SWIFT | 0.14 | [0.12, 0.16] |
| | | Train-label-dist | −0.08 | [−0.09, −0.06] |
| | | Uniform | −0.25 | [−0.29, −0.22] |
| | FFC | Human | 0.21 | [0.20, 0.22] |
| | | SCANDL | 0.16 | [0.14, 0.17] |
| | | Eyettention | 0.09 | [0.07, 0.10] |
| | | E-Z reader | 0.13 | [0.11, 0.15] |
| | | SWIFT | 0.14 | [0.12, 0.16] |
| | | Train-label-dist | −0.07 | [−0.09, −0.05] |
| | | Uniform | −0.23 | [−0.27, −0.19] |

Table 7: Posterior distributions of effect sizes for three psycholinguistic predictors—lexical frequency, surprisal and word length—in the *New Reader/New Sentence* setting. Estimates were computed via Bayesian generalized linear models. We report means and 95%-credible intervals $[2.5\%; 97.5\%]$ for three psycholinguistic effects on four reading measures extracted from the fixation sequences.

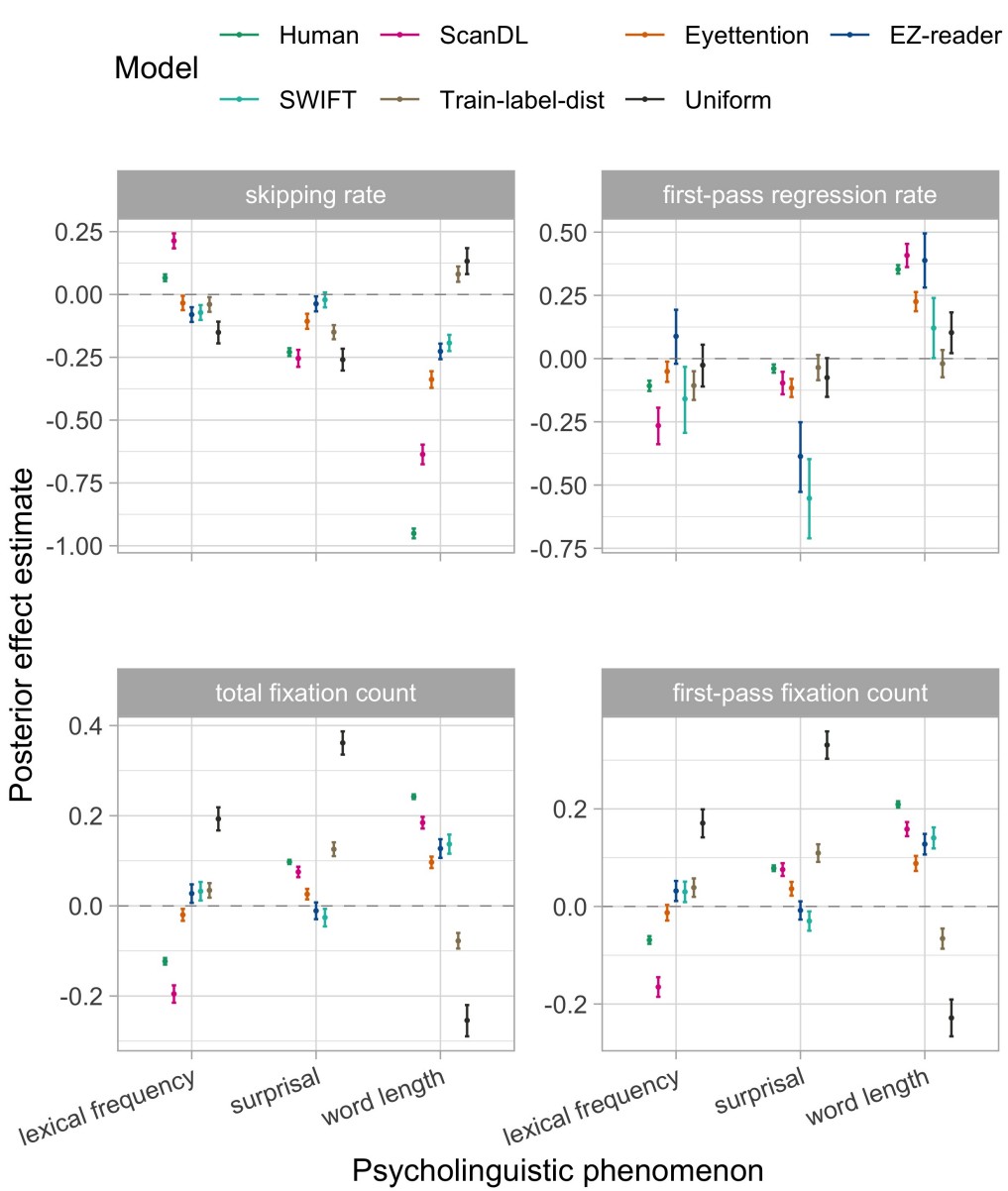

Figure 8: Comparison of posterior effect estimates for psycholinguistic phenomena on reading measures between original and predicted scanpaths. Lines represent a 95% credible interval, means are denoted by dots.

## E   Emulation of Reading Pattern Variability

| Reading measure | Definition |
|---|---|
| Regression rate | The average probability of a word in the sentence to be the starting point of a regression |
| Normalized fixation count | The average number of fixations in a sentence |
| Progressive saccade length | The average length of progressive saccades |
| Regressive saccade length | The average length of regressive saccades |
| Skipping rate | The average probability of a word to be skipped |
| First-pass count | The average number of fixations on a word during the first-pass reading of the sentence |

Table 8: Definition of reading measures used for the investigation of the reading pattern predictability across readers and the predictability patterns across models.

| Reading measure | CELER | | ZuCo | |
|---|---|---|---|---|
| | mean ± sd (true) | mean ± sd (SCANDL) | mean ± sd (true) | mean ± sd (SCANDL) |
| First pass count | 0.91 ± 0.21 | 0.90 ± 0.19 | 0.84 ± 0.21 | 0.73 ± 0.25 |
| Normalized fixation count | 1.24 ± 0.50 | 1.03 ± 0.30 | 1.15 ± 0.45 | 0.83 ± 0.34 |
| Progr. saccade length | 2.00 ± 0.60 | 1.75 ± 0.63 | 2.72 ± 1.06 | 2.00 ± 0.60 |
| Regr. saccade length | 2.55 ± 2.16 | 1.82 ± 2.24 | 4.44 ± 3.58 | 2.44 ± 3.05 |
| Skipping rate | 0.29 ± 0.13 | 0.27 ± 0.12 | 0.42 ± 0.16 | 0.41 ± 0.18 |
| Regression rate | 0.14 ± 0.10 | 0.08 ± 0.07 | 0.15 ± 0.09 | 0.07 ± 0.07 |

Table 9: Mean and standard deviation of the reading measures of true and predicted scanpaths of both the CELER (Berzak et al., 2022) and ZuCo (Hollenstein et al., 2018) datasets.

| Reading measure | CELER | ZuCo |
|---|---|---|
| First-pass fixation count | .08 | -.39 |
| Normalized fixation count | .52*** | .07 |
| Progr. saccade length | .48*** | .77** |
| Regr. saccade length | .32** | .04 |
| Skipping rate | .22 | .70* |
| Regression rate | .62*** | .75** |

Table 10: Pearson correlations between reader's mean NLD and mean reading measures. Statistically significant correlations are marked with an asterisk. ***) $p \leq 0.001$, **) $p \leq 0.01$, *) $p \leq 0.05$.

| Baseline | Pearson corr. coef. |
|---|---|
| Eyettention | .41*** |
| Uniform | .41*** |
| Traindist | .39*** |
| E-Z Reader | .37*** |
| SWIFT | .36*** |

Table 11: Pearson correlations between the mean sentence NLDs of SCANDL and every baseline model. Statistically significant correlations are marked with an asterisk. ***) $p \leq 0.001$,**) $p \leq 0.01$,*) $p \leq 0.05$.

## F   Model-Specific Predictability Patterns

| Model | Reading measure | Pearson corr. |
|---|---|---|
| SCANDL | Normalized fixation count | 0.38*** |
| | Regression rate | 0.31*** |
| | Progressive saccade length | 0.31*** |
| | Skipping rate | 0.24*** |
| | Regressive saccade length | 0.22*** |
| | First-pass fixation count | 0.03 |
| Eyettention | Progressive saccade length | 0.31*** |
| | Normalized fixation count | 0.25*** |
| | Regression rate | 0.19*** |
| | Skipping rate | 0.18*** |
| | Regressive saccade length | 0.18*** |
| | First-pass fixation count | 0.02 |
| E-Z reader | Normalized fixation count | 0.69*** |
| | First-pass fixation count | 0.46*** |
| | Regression rate | 0.45*** |
| | Regressive saccade length | 0.16*** |
| | Progressive saccade length | 0.01 |
| | Skipping rate | −0.2*** |
| SWIFT | Normalized fixation count | 0.71*** |
| | First-pass fixation count | 0.49*** |
| | Regression rate | 0.47*** |
| | Regressive saccade length | 0.16*** |
| | Progressive saccade length | −0.02 |
| | Skipping rate | −0.24*** |
| Traindist | Normalized fixation count | 0.33*** |
| | Regression rate | 0.22*** |
| | Progressive saccade length | 0.2*** |
| | Regressive saccade length | 0.2*** |
| | First-pass fixation count | 0.16*** |
| | Skipping rate | 0.01 |
| Uniform | Normalized fixation count | 0.55*** |
| | First-pass fixation count | 0.41*** |
| | Regression rate | 0.37*** |
| | Regressive saccade length | 0.23*** |
| | Progressive saccade length | 0.03 |
| | Skipping rate | −0.23*** |

Table 12: Pearson correlations between the NLD of a predicted scanpath and reading measures of the corresponding true scanpath, for SCANDL and the baselines. Statistically significant correlations are marked with an asterisk. ***) $p \leq 0.001$,**) $p \leq 0.01$,*) $p \leq 0.05$.

## G Derivation of Objective

In this section we derive the component $\mathcal{L}_{\text{VLB}}$ (Equation 1) of the model's training objective $\mathcal{L}_{\text{SCANDL}}$ from the variational lower bound (Equation 2). This derivation closely follows the one given by Luo (2022).

The diffusion posterior is defined as

$$q(\mathbf{z}_{1:T}|\mathbf{z}_0) = \prod_{t=1}^{T} q(\mathbf{z}_t|\mathbf{z}_{t-1}). \tag{4}$$

Each forward transition in the noising process follows a first-order Markov property by being dependent only on the immediately preceding latent, and it is written as

$$q(\mathbf{z}_t|\mathbf{z}_{t-1}) = \mathcal{N}(\sqrt{\alpha_t}\mathbf{z}_{t-1}, (1-\alpha_t)\mathbb{I}). \tag{5}$$

The joint distribution of all latents of a diffusion model is then given by

$$p(\mathbf{z}_{0:T}) = p(\mathbf{z}_T) \prod_{t=1}^{T} p_\theta(\mathbf{z}_{t-1}|\mathbf{z}_t), \tag{6}$$

where $p(\mathbf{z}_T) = \mathcal{N}(0, \mathbb{I})$.

A diffusion model can be optimized by maximizing the log-likelihood of the data $\log p(\mathbf{z}_0)$, which is equivalent to maximizing the variational lower bound (VLB):

$$\log p(\mathbf{z}_0) = \log \int p(\mathbf{z}_{0:T}) \, d\mathbf{z}_{1:T} \tag{7}$$

$$= \log \int \frac{p(\mathbf{z}_{0:T}) \, q(\mathbf{z}_{1:T} \mid \mathbf{z}_0)}{q(\mathbf{z}_{1:T} \mid \mathbf{z}_0)} d\mathbf{z}_{1:T} \tag{8}$$

$$= \log \mathbb{E}_{q(\mathbf{z}_{1:T} \mid \mathbf{z}_0)} \left[ \frac{p(\mathbf{z}_{0:T})}{q(\mathbf{z}_{1:T} \mid \mathbf{z}_0)} \right] \tag{9}$$

$$\geq \mathbb{E}_{q(\mathbf{z}_{1:T} \mid \mathbf{z}_0)} \left[ \log \frac{p(\mathbf{z}_{0:T})}{q(\mathbf{z}_{1:T} \mid \mathbf{z}_0)} \right] \text{(Jensen's Inequality)} \tag{10}$$

$$= \mathbb{E}_{q(\mathbf{z}_{1:T} \mid \mathbf{z}_0)} \left[ \log \frac{p(\mathbf{z}_T) \prod_{t=1}^{T} p_{\boldsymbol{\theta}}(\mathbf{z}_{t-1} \mid \mathbf{z}_t)}{\prod_{t=1}^{T} q(\mathbf{z}_t \mid \mathbf{z}_{t-1})} \right] \text{Equations 4, 6} \tag{11}$$

$$= \mathbb{E}_{q(\mathbf{z}_{1:T} \mid \mathbf{z}_0)} \left[ \log \frac{p(\mathbf{z}_T) \, p_{\boldsymbol{\theta}}(\mathbf{z}_0 \mid \mathbf{z}_1) \prod_{t=2}^{T} p_{\boldsymbol{\theta}}(\mathbf{z}_{t-1} \mid \mathbf{z}_t)}{q(\mathbf{z}_1 \mid \mathbf{z}_0) \prod_{t=2}^{T} q(\mathbf{z}_t \mid \mathbf{z}_{t-1})} \right] \tag{12}$$

$$= \mathbb{E}_{q(\mathbf{z}_{1:T} \mid \mathbf{z}_0)} \left[ \log \frac{p_{\boldsymbol{\theta}}(\mathbf{z}_T) \, p_{\boldsymbol{\theta}}(\mathbf{z}_0 \mid \mathbf{z}_1)}{q(\mathbf{z}_1 \mid \mathbf{z}_0)} + \log \prod_{t=2}^{T} \frac{p_{\boldsymbol{\theta}}(\mathbf{z}_{t-1} \mid \mathbf{z}_t)}{q(\mathbf{z}_t \mid \mathbf{z}_{t-1}, \mathbf{z}_0)} \right] \tag{13}$$

$$= \mathbb{E}_{q(\mathbf{x}_{1:T} \mid \mathbf{x}_0)} \left[ \log \frac{p(\mathbf{x}_T) \, p_{\boldsymbol{\theta}}(\mathbf{x}_0 \mid \mathbf{x}_1)}{q(\mathbf{x}_1 \mid \mathbf{x}_0)} + \log \prod_{t=2}^{T} \frac{p_{\boldsymbol{\theta}}(\mathbf{x}_{t-1} \mid \mathbf{x}_t)}{\frac{q(\mathbf{x}_{t-1} \mid \mathbf{x}_t, \mathbf{x}_0) q(\mathbf{x}_t \mid \mathbf{x}_0)}{q(\mathbf{x}_{t-1} \mid \mathbf{x}_0)}} \right] \tag{14}$$

$$= \mathbb{E}_{q(\mathbf{z}_{1:T} \mid \mathbf{z}_0)} \left[ \log \frac{p(\mathbf{z}_T) \, p_{\boldsymbol{\theta}}(\mathbf{z}_0 \mid \mathbf{z}_1)}{q(\mathbf{z}_1 \dagger \mathbf{z}_0)} + \log \frac{q(\mathbf{z}_1 \dagger \mathbf{z}_0)}{q(\mathbf{z}_T \mid \mathbf{z}_0)} + \log \prod_{t=2}^{T} \frac{p_{\boldsymbol{\theta}}(\mathbf{z}_{t-1} \mid \mathbf{z}_t)}{q(\mathbf{z}_{t-1} \mid \mathbf{z}_t, \mathbf{z}_0)} \right] \tag{15}$$

$$= \mathbb{E}_{q(\mathbf{z}_{1:T} \mid \mathbf{z}_0)} \left[ \log \frac{p(\mathbf{z}_T) \, p_{\boldsymbol{\theta}}(\mathbf{z}_0 \mid \mathbf{z}_1)}{q(\mathbf{z}_T \mid \mathbf{z}_0)} + \sum_{t=2}^{T} \log \frac{p_{\boldsymbol{\theta}}(\mathbf{z}_{t-1} \mid \mathbf{z}_t)}{q(\mathbf{z}_{t-1} \mid \mathbf{z}_t, \mathbf{z}_0)} \right] \tag{16}$$

$$= \mathbb{E}_{q(\mathbf{z}_{1:T} \mid \mathbf{z}_0)} \left[ \log p_{\boldsymbol{\theta}}(\mathbf{z}_0 \mid \mathbf{z}_1) \right] + \mathbb{E}_{q(\mathbf{z}_{1:T} \mid \mathbf{z}_0)} \left[ \log \frac{p(\mathbf{z}_T)}{q(\mathbf{z}_T \mid \mathbf{z}_0)} \right] + \sum_{t=2}^{T} \mathbb{E}_{q(\mathbf{z}_{1:T} \mid \mathbf{z}_0)} \left[ \log \frac{p_{\boldsymbol{\theta}}(\mathbf{z}_{t-1} \mid \mathbf{z}_t)}{q(\mathbf{z}_{t-1} \mid \mathbf{z}_t, \mathbf{z}_0)} \right] \tag{17}$$

$$= \mathbb{E}_{q(\mathbf{z}_1 \mid \mathbf{z}_0)} \left[ \log p_{\boldsymbol{\theta}}(\mathbf{z}_0 \mid \mathbf{z}_1) \right] + \mathbb{E}_{q(\mathbf{z}_T \mid \mathbf{z}_0)} \left[ \log \frac{p(\mathbf{z}_T)}{q(\mathbf{z}_T \mid \mathbf{z}_0)} \right] + \sum_{t=2}^{T} \mathbb{E}_{q(\mathbf{z}_t, \mathbf{z}_{t-1} \mid \mathbf{z}_0)} \left[ \log \frac{p_{\boldsymbol{\theta}}(\mathbf{z}_{t-1} \mid \mathbf{z}_t)}{q(\mathbf{z}_{t-1} \mid \mathbf{z}_t, \mathbf{z}_0)} \right] \tag{18}$$

$$= \underbrace{\mathbb{E}_{q(\mathbf{z}_1 \mid \mathbf{z}_0)} \left[ \log p_{\boldsymbol{\theta}}(\mathbf{z}_0 \mid \mathbf{z}_1) \right]}_{\text{reconstruction term}} - \underbrace{D_{\mathrm{KL}} \left( q(\mathbf{z}_T \mid \mathbf{z}_0) \, \| \, p(\mathbf{z}_T) \right)}_{\text{prior matching term}} - \tag{19}$$

$$\sum_{t=2}^{T} \underbrace{\mathbb{E}_{q(\mathbf{z}_t \mid \mathbf{z}_0)} \left[ D_{\mathrm{KL}} \left( q(\mathbf{z}_{t-1} \mid \mathbf{z}_t, \mathbf{z}_0) \, \| \, p_{\boldsymbol{\theta}}(\mathbf{z}_{t-1} \mid \mathbf{z}_t) \right) \right]]}_{\text{denoising matching term}} \tag{20}$$

The *denoising matching term* contains KL Divergence terms (Csiszar, 1975) between the true denoising transition $q(\mathbf{z}_{t-1} \mid \mathbf{z}_t, \mathbf{z}_0)$ and the approximated transition $p_{\boldsymbol{\theta}}(\mathbf{z}_{t-1} \mid \mathbf{z}_t)$. For our learned distribution $p_{\boldsymbol{\theta}}$ to match the ground-truth distribution $q$ as closely as possible, the *denoising matching term* has to be minimized. However, the summation term has a high optimization cost, which is why we make optimization tractable by leveraging the Gaussian transition via Bayes rule:

$$q(\mathbf{z}_{t-1} \mid \mathbf{z}_t, \mathbf{z}_0) = \frac{q(\mathbf{z}_t \mid \mathbf{z}_{t-1}, \mathbf{z}_0) \, q(\mathbf{z}_{t-1} \mid \mathbf{z}_0)}{q(\mathbf{z}_t \mid \mathbf{z}_0)}, \tag{21}$$

where $q(\mathbf{z}_t \mid \mathbf{z}_{t-1}, \mathbf{z}_0) = q(\mathbf{z}_t \mid \mathbf{z}_{t-1}) = \mathcal{N}\left(\sqrt{\alpha_t}\mathbf{z}_{t-1}, (1 - \alpha_t)\,\mathbb{I}\right)$ and, applying the reparametrization trick, $\mathbf{z}_t = \sqrt{\alpha_t}\mathbf{z}_{t-1} + \sqrt{1 - \alpha_t}\epsilon$ with $\epsilon \sim \mathcal{N}(\mathbf{0}, \mathbb{I})$ and $\mathbf{z}_{t-1} = \sqrt{\alpha_{t-1}}\mathbf{z}_{t-2} + \sqrt{1 - \alpha_{t-1}}\epsilon$ with $\epsilon \sim \mathcal{N}(\mathbf{0}, \mathbb{I})$. The form of $q(\mathbf{z}_t \mid \mathbf{z}_0)$ is derived by repeatedly applying the reparametrization trick. Let there be $2T$ noise variables $\{\epsilon_t^*, \epsilon_t\}_{t=0}^{T} \overset{\text{iid}}{\sim} \mathcal{N}(\mathbf{0}, \mathbf{I})$. A sample $\mathbf{z}_t \sim q(\mathbf{z}_t \mid \mathbf{z}_0)$ can be re-written as:

$$\mathbf{z}_t = \sqrt{\alpha_t}\mathbf{z}_{t-1} + \sqrt{1-\alpha_t}\epsilon^*_{t-1} \tag{22}$$

$$= \sqrt{\alpha_t}\left(\sqrt{\alpha_{t-1}}\mathbf{z}_{t-2} + \sqrt{1-\alpha_{t-1}}\epsilon^*_{t-2}\right) + \sqrt{1-\alpha_t}\epsilon^*_{t-1} \tag{23}$$

$$= \sqrt{\alpha_t\alpha_{t-1}}\mathbf{z}_{t-2} + \sqrt{\alpha_t - \alpha_t\alpha_{t-1}}\epsilon^*_{t-2} + \sqrt{1-\alpha_t}\epsilon^*_{t-1} \tag{24}$$

$$= \sqrt{\alpha_t\alpha_{t-1}}\mathbf{z}_{t-2} + \sqrt{\sqrt{\alpha_t - \alpha_t\alpha_{t-1}}^2 + \sqrt{1-\alpha_t^2}}\epsilon_{t-2} \tag{25}$$

$$= \sqrt{\alpha_t\alpha_{t-1}}\mathbf{z}_{t-2} + \sqrt{\alpha_t - \alpha_t\alpha_{t-1} + 1 - \alpha_t}\epsilon_{t-2} \tag{26}$$

$$= \sqrt{\alpha_t\alpha_{t-1}}\mathbf{z}_{t-2} + \sqrt{1-\alpha_t\alpha_{t-1}}\epsilon_{t-2} \tag{27}$$

$$= \ldots \tag{28}$$

$$= \sqrt{\prod_{i=1}^{t}\alpha_i}\mathbf{z}_0 + \sqrt{1-\prod_{i=1}^{t}\alpha_i}\epsilon_0 \tag{29}$$

$$= \sqrt{\bar{\alpha}_t}\mathbf{z}_0 + \sqrt{1-\bar{\alpha}_t}\epsilon_0 \tag{30}$$

$$\sim \mathcal{N}\left(\sqrt{\bar{\alpha}_t}\mathbf{z}_0, (1-\bar{\alpha}_t)\mathbb{I}\right) \tag{31}$$

$q(\mathbf{z}_{t-1} \mid \mathbf{z}_t, \mathbf{z}_0)$ can now be obtained by substituting into the Bayes' expansion from Equation 21:

$$q\left(\mathbf{z}_{t-1} \mid \mathbf{z}_t, \mathbf{z}_0\right) = \frac{q\left(\mathbf{z}_t \mid \mathbf{z}_{t-1}, \mathbf{z}_0\right) q\left(\mathbf{z}_{t-1} \mid \mathbf{z}_0\right)}{q\left(\mathbf{z}_t \mid \mathbf{z}_0\right)} \tag{32}$$

$$= \frac{\mathcal{N}\left(\sqrt{\alpha_t}\mathbf{z}_{t-1}, \left(1-\alpha_t\right)\mathbb{I}\right) \mathcal{N}\left(\sqrt{\bar{\alpha}_{t-1}}\mathbf{z}_0, \left(1-\bar{\alpha}_{t-1}\right)\mathbb{I}\right)}{\mathcal{N}\left(\sqrt{\bar{\alpha}_t}\mathbf{z}_0, \left(1-\bar{\alpha}_t\right)\mathbb{I}\right)} \tag{33}$$

$$\propto \exp\left\{-\left[\frac{\left(\mathbf{z}_t - \sqrt{\alpha_t}\mathbf{z}_{t-1}\right)^2}{2\left(1-\alpha_t\right)} + \frac{\left(\mathbf{z}_{t-1} - \sqrt{\bar{\alpha}_{t-1}}\mathbf{z}_0\right)^2}{2\left(1-\bar{\alpha}_{t-1}\right)} - \frac{\left(\mathbf{z}_t - \sqrt{\bar{\alpha}_t}\mathbf{z}_0\right)^2}{2\left(1-\bar{\alpha}_t\right)}\right]\right\} \tag{34}$$

$$= \exp\left\{-\frac{1}{2}\left[\frac{\left(\mathbf{z}_t - \sqrt{\alpha_t}\mathbf{z}_{t-1}\right)^2}{1-\alpha_t} + \frac{\left(\mathbf{z}_{t-1} - \sqrt{\bar{\alpha}_{t-1}}\mathbf{z}_0\right)^2}{1-\bar{\alpha}_{t-1}} - \frac{\left(\mathbf{z}_t - \sqrt{\bar{\alpha}_t}\mathbf{z}_0\right)^2}{1-\bar{\alpha}_t}\right]\right\} \tag{35}$$

$$= \exp\left\{-\frac{1}{2}\left[\frac{\left(-2\sqrt{\alpha_t}\mathbf{z}_t\mathbf{z}_{t-1} + \alpha_t\mathbf{z}_{t-1}^2\right)}{1-\alpha_t} + \frac{\left(\mathbf{z}_{t-1}^2 - 2\sqrt{\bar{\alpha}_{t-1}}\mathbf{z}_{t-1}\mathbf{z}_0\right)}{1-\bar{\alpha}_{t-1}} + C\left(\mathbf{z}_t, \mathbf{z}_0\right)\right]\right\} \tag{36}$$

$$\propto \exp\left\{-\frac{1}{2}\left[-\frac{2\sqrt{\alpha_t}\mathbf{z}_t\mathbf{z}_{t-1}}{1-\alpha_t} + \frac{\alpha_t\mathbf{z}_{t-1}^2}{1-\alpha_t} + \frac{\mathbf{z}_{t-1}^2}{1-\bar{\alpha}_{t-1}} - \frac{2\sqrt{\bar{\alpha}_{t-1}}\mathbf{z}_{t-1}\mathbf{z}_0}{1-\bar{\alpha}_{t-1}}\right]\right\} \tag{37}$$

$$= \exp\left\{-\frac{1}{2}\left[\left(\frac{\alpha_t}{1-\alpha_t} + \frac{1}{1-\bar{\alpha}_{t-1}}\right)\mathbf{z}_{t-1}^2 - 2\left(\frac{\sqrt{\alpha_t}\mathbf{z}_t}{1-\alpha_t} + \frac{\sqrt{\bar{\alpha}_{t-1}}\mathbf{z}_0}{1-\bar{\alpha}_{t-1}}\right)\mathbf{z}_{t-1}\right]\right\} \tag{38}$$

$$= \exp\left\{-\frac{1}{2}\left[\frac{\alpha_t\left(1-\bar{\alpha}_{t-1}\right) + 1-\alpha_t}{\left(1-\alpha_t\right)\left(1-\bar{\alpha}_{t-1}\right)}\mathbf{z}_{t-1}^2 - 2\left(\frac{\sqrt{\alpha_t}\mathbf{z}_t}{1-\alpha_t} + \frac{\sqrt{\bar{\alpha}_{t-1}}\mathbf{z}_0}{1-\bar{\alpha}_{t-1}}\right)\mathbf{z}_{t-1}\right]\right\} \tag{39}$$

$$= \exp\left\{-\frac{1}{2}\left[\frac{\alpha_t - \bar{\alpha}_t + 1-\alpha_t}{\left(1-\alpha_t\right)\left(1-\bar{\alpha}_{t-1}\right)}\mathbf{z}_{t-1}^2 - 2\left(\frac{\sqrt{\alpha_t}\mathbf{z}_t}{1-\alpha_t} + \frac{\sqrt{\bar{\alpha}_{t-1}}\mathbf{z}_0}{1-\bar{\alpha}_{t-1}}\right)\mathbf{z}_{t-1}\right]\right\} \tag{40}$$

$$= \exp\left\{-\frac{1}{2}\left[\frac{1-\bar{\alpha}_t}{\left(1-\alpha_t\right)\left(1-\bar{\alpha}_{t-1}\right)}\mathbf{z}_{t-1}^2 - 2\left(\frac{\sqrt{\alpha_t}\mathbf{z}_t}{1-\alpha_t} + \frac{\sqrt{\bar{\alpha}_{t-1}}\mathbf{z}_0}{1-\bar{\alpha}_{t-1}}\right)\mathbf{z}_{t-1}\right]\right\} \tag{41}$$

$$= \exp\left\{-\frac{1}{2}\left(\frac{1-\bar{\alpha}_t}{\left(1-\alpha_t\right)\left(1-\bar{\alpha}_{t-1}\right)}\right)\left[\mathbf{z}_{t-1}^2 - 2\frac{\left(\frac{\sqrt{\alpha_t}\mathbf{z}_t}{1-\alpha_t} + \frac{\sqrt{\bar{\alpha}_{t-1}}\mathbf{z}_0}{1-\bar{\alpha}_{t-1}}\right)}{\frac{1-\bar{\alpha}_t}{\left(1-\alpha_t\right)\left(1-\bar{\alpha}_{t-1}\right)}}\mathbf{z}_{t-1}\right]\right\} \tag{42}$$

$$= \exp\left\{-\frac{1}{2}\left(\frac{1-\bar{\alpha}_t}{\left(1-\alpha_t\right)\left(1-\bar{\alpha}_{t-1}\right)}\right)\left[\mathbf{z}_{t-1}^2 - 2\frac{\left(\frac{\sqrt{\alpha_t}\mathbf{z}_t}{1-\alpha_t} + \frac{\sqrt{\bar{\alpha}_{t-1}}\mathbf{z}_0}{1-\bar{\alpha}_{t-1}}\right)\left(1-\alpha_t\right)\left(1-\bar{\alpha}_{t-1}\right)}{1-\bar{\alpha}_t}\mathbf{z}_{t-1}\right]\right\} \tag{43}$$

$$= \exp\left\{-\frac{1}{2}\left(\frac{1}{\frac{\left(1-\alpha_t\right)\left(1-\bar{\alpha}_{t-1}\right)}{1-\bar{\alpha}_t}}\right)\left[\mathbf{z}_{t-1}^2 - 2\frac{\sqrt{\alpha_t}\left(1-\bar{\alpha}_{t-1}\right)\mathbf{z}_t + \sqrt{\bar{\alpha}_{t-1}}\left(1-\alpha_t\right)\mathbf{z}_0}{1-\bar{\alpha}_t}\mathbf{z}_{t-1}\right]\right\} \tag{44}$$

$$\propto \mathcal{N}(\underbrace{\frac{\sqrt{\alpha_t}\left(1-\bar{\alpha}_{t-1}\right)\mathbf{z}_t + \sqrt{\bar{\alpha}_{t-1}}\left(1-\alpha_t\right)\mathbf{z}_0}{1-\bar{\alpha}_t}}_{\mu_q(\mathbf{z}_t, \mathbf{z}_0)}, \underbrace{\frac{\left(1-\alpha_t\right)\left(1-\bar{\alpha}_{t-1}\right)}{1-\bar{\alpha}_t}\mathbb{I}}_{\Sigma_q(t)}) \tag{45}$$

$$\tag{46}$$

Therefore, each step $\mathbf{z}_{t-1} \sim q\left(\mathbf{z}_{t-1} \mid \mathbf{z}_t, \mathbf{z}_0\right)$ is normally distributed, with mean $\mu_q\left(\mathbf{z}_t, \mathbf{z}_0\right)$ and variance $\Sigma_q(t)$. Following Equation 45, the variance can be written as $\sigma_q^2(t)\mathbb{I}$, where

$$\sigma_q^2(t) = \frac{\left(1-\alpha_t\right)\left(1-\bar{\alpha}_{t-1}\right)}{1-\bar{\alpha}_t}. \tag{47}$$

As all $\alpha$ terms are given a priori by the noise schedule, the variance can be immediately computed. The mean, however, must be parametrized, as it is a function of $\mathbf{z}_t$, hence $\mu_{\boldsymbol{\theta}}\left(\mathbf{z}_t, t\right)$. Since we choose the variances of the two distributions to match exactly, we can optimize the KL Divergence by minimizing the difference between the means of the two distributions:

$$\operatorname*{argmin}_{\boldsymbol{\theta}} D_{\mathrm{KL}}\left(q\left(\mathbf{z}_{t-1} \mid \mathbf{z}_t, \mathbf{z}_0\right) \| p_{\boldsymbol{\theta}}\left(\mathbf{z}_{t-1} \mid \mathbf{z}_t\right)\right) \tag{48}$$

$$= \operatorname*{argmin}_{\boldsymbol{\theta}} D_{\mathrm{KL}}\left(\mathcal{N}\left(\mathbf{z}_{t-1}; \mu_q, \boldsymbol{\Sigma}_q(t)\right) \| \mathcal{N}\left(\mu_{\boldsymbol{\theta}}, \boldsymbol{\Sigma}_q(t)\right)\right) \tag{49}$$

$$= \operatorname*{argmin}_{\boldsymbol{\theta}} \frac{1}{2}\left[\log \frac{|\boldsymbol{\Sigma}_q(t)|}{|\boldsymbol{\Sigma}_q(t)|} - d + \operatorname{tr}\left(\boldsymbol{\Sigma}_q(t)^{-1}\boldsymbol{\Sigma}_q(t)\right) + \left(\mu_{\boldsymbol{\theta}} - \mu_q\right)^T \boldsymbol{\Sigma}_q(t)^{-1}\left(\mu_{\boldsymbol{\theta}} - \mu_q\right)\right] \tag{50}$$

$$= \operatorname*{argmin}_{\boldsymbol{\theta}} \frac{1}{2}\left[\log 1 - d + d + \left(\mu_{\boldsymbol{\theta}} - \mu_q\right)^T \boldsymbol{\Sigma}_q(t)^{-1}\left(\mu_{\boldsymbol{\theta}} - \mu_q\right)\right] \tag{51}$$

$$= \operatorname*{argmin}_{\boldsymbol{\theta}} \frac{1}{2}\left[\left(\mu_{\boldsymbol{\theta}} - \mu_q\right)^T \boldsymbol{\Sigma}_q(t)^{-1}\left(\mu_{\boldsymbol{\theta}} - \mu_q\right)\right] \tag{52}$$

$$= \operatorname*{argmin}_{\boldsymbol{\theta}} \frac{1}{2}\left[\left(\mu_{\boldsymbol{\theta}} - \mu_q\right)^T \left(\sigma_q^2(t)\mathbb{I}\right)^{-1}\left(\mu_{\boldsymbol{\theta}} - \mu_q\right)\right] \tag{53}$$

$$= \operatorname*{argmin}_{\boldsymbol{\theta}} \frac{1}{2\sigma_q^2(t)}\left[\|\mu_{\boldsymbol{\theta}} - \mu_q\|_2^2\right] \tag{54}$$

This means that we want to optimize $\mu_{\boldsymbol{\theta}}\left(\mathbf{z}_t, t\right)$ so that it matches $\mu_q\left(\mathbf{z}_t, \mathbf{z}_0\right)$, which is defined as

$$\mu_q\left(\mathbf{z}_t, \mathbf{z}_0\right) = \frac{\sqrt{\alpha_t}\left(1 - \bar{\alpha}_{t-1}\right)\mathbf{z}_t + \sqrt{\bar{\alpha}_{t-1}}\left(1 - \alpha_t\right)\mathbf{z}_0}{1 - \bar{\alpha}_t}, \tag{55}$$

and $\mu_{\boldsymbol{\theta}}\left(\mathbf{z}_t, t\right)$ is defined as

$$\mu_{\boldsymbol{\theta}}\left(\mathbf{z}_t, t\right) = \frac{\sqrt{\alpha_t}\left(1 - \bar{\alpha}_{t-1}\right)\mathbf{z}_t + \sqrt{\bar{\alpha}_{t-1}}\left(1 - \alpha_t\right)\hat{\mathbf{z}}_{\boldsymbol{\theta}}\left(\mathbf{z}_t, t\right)}{1 - \bar{\alpha}_t}, \tag{56}$$

where $\hat{\mathbf{z}}_{\boldsymbol{\theta}}\left(\mathbf{z}_t, t\right)$ is parametrized by our transformer model $f_{\boldsymbol{\theta}}$ that is trained to predict $\mathbf{z}_0$ from $\mathbf{z}_t$. We henceforth denote $\hat{\mathbf{z}}_{\boldsymbol{\theta}}\left(\mathbf{z}_t, t\right)$ by $f_{\boldsymbol{\theta}}\left(\mathbf{z}_t, t\right)$, where $f_{\boldsymbol{\theta}}$ is our transformer model and $f_{\boldsymbol{\theta}}\left(\mathbf{z}_t, t\right)$ is the model prediction $\hat{\mathbf{z}_0}$ for ground-truth $\mathbf{z}_0$. The objective is then simplified to:

$$\operatorname*{argmin}_{\boldsymbol{\theta}} D_{\mathrm{KL}}\left(q\left(\mathbf{z}_{t-1} \mid \mathbf{z}_t, \mathbf{z}_0\right) \| p_{\boldsymbol{\theta}}\left(\mathbf{z}_{t-1} \mid \mathbf{z}_t\right)\right) \tag{57}$$

$$= \operatorname*{argmin}_{\boldsymbol{\theta}} D_{\mathrm{KL}}\left(\mathcal{N}\left(\mu_q, \boldsymbol{\Sigma}_q(t)\right) \| \mathcal{N}\left(\mu_{\boldsymbol{\theta}}, \boldsymbol{\Sigma}_q(t)\right)\right) \tag{58}$$

$$= \operatorname*{argmin}_{\boldsymbol{\theta}} \frac{1}{2\sigma_q^2(t)}\left[\left\|\frac{\sqrt{\alpha_t}\left(1 - \bar{\alpha}_{t-1}\right)\mathbf{z}_t + \sqrt{\bar{\alpha}_{t-1}}\left(1 - \alpha_t\right)f_{\boldsymbol{\theta}}\left(\mathbf{z}_t, t\right)}{1 - \bar{\alpha}_t} - \frac{\sqrt{\alpha_t}\left(1 - \bar{\alpha}_{t-1}\right)\mathbf{z}_t + \sqrt{\bar{\alpha}_{t-1}}\left(1 - \alpha_t\right)\mathbf{z}_0}{1 - \bar{\alpha}_t}\right\|_2^2\right] \tag{59}$$

$$= \operatorname*{argmin}_{\boldsymbol{\theta}} \frac{1}{2\sigma_q^2(t)}\left[\left\|\frac{\sqrt{\bar{\alpha}_{t-1}}\left(1 - \alpha_t\right)f_{\boldsymbol{\theta}}\left(\mathbf{z}_t, t\right)}{1 - \bar{\alpha}_t} - \frac{\sqrt{\bar{\alpha}_{t-1}}\left(1 - \alpha_t\right)\mathbf{z}_0}{1 - \bar{\alpha}_t}\right\|_2^2\right] \tag{60}$$

$$= \operatorname*{argmin}_{\boldsymbol{\theta}} \frac{1}{2\sigma_q^2(t)}\left[\left\|\frac{\sqrt{\bar{\alpha}_{t-1}}\left(1 - \alpha_t\right)}{1 - \bar{\alpha}_t}\left(f_{\boldsymbol{\theta}}\left(\mathbf{z}_t, t\right) - \mathbf{z}_0\right)\right\|_2^2\right] \tag{61}$$

$$= \operatorname*{argmin}_{\boldsymbol{\theta}} \frac{1}{2\sigma_q^2(t)}\frac{\bar{\alpha}_{t-1}\left(1 - \alpha_t\right)^2}{\left(1 - \bar{\alpha}_t\right)^2}\left[\|f_{\boldsymbol{\theta}}\left(\mathbf{z}_t, t\right) - \mathbf{z}_0\|_2^2\right] \tag{62}$$

This means that maximizing the VLB can be achieved by learning a neural network that predicts the ground truth from the noised version of the ground truth. In our case, we neglect the constant term and denote the part of our optimization criterion that stems from the VLB as

$$\mathcal{L}_{\mathrm{VLB}} = \|f_{\boldsymbol{\theta}}\left(\mathbf{z}_t, t\right) - \mathbf{z}_0\|_2^2.$$