# OpenReview forum: "ScanDL: A Diffusion Model for Generating Synthetic Scanpaths on Texts"
_EMNLP/2023/Conference — EMNLP 2023 Main_

### Official Review · Reviewer_iKVr · 2023-08-03

**Soundness:** 5

**Excitement:**

4: Strong: This paper deepens the understanding of some phenomenon or lowers the barriers to an existing research direction.

**Paper Topic And Main Contributions:**

This paper proposes a model to generate synthetic eye-movements on text. Based on a
discrete diffusion model with appropriate tweaks, the authors succeed in generating
eye-movements that is even better than human performance. While the main contribution is
methodological, experimental investigation is also sufficient to support the main claim.

This paper is interesting and will be useful for many disciplines. Since gaze information
is difficult to obtain experimentally, this kind of synthetic eye-movements are
beneficial for many problems with cognitive effects, such as readability, grammaticality,
sentiment analysis, and so on. Improvements from previous state-of-the-art seems to be
significant, thus this method could be a new standard on this task.

With respect to its performance, the only regret is a lack of qualitative comparison
with previous methods. Using diffusion models, *how* the ScanDL performs different from
previous models? Where is it strong, and how it performs diffenrently from human?
Such a qualitative information will further strengthen the arguments to adopt a diffusion
model on this kind of problem.

**Reasons To Accept:**

Establishes a new state-of-the-art on generating eye movements that has human-level performance. Methodological contribution is also strong.

**Reasons To Reject:**

Not especially.

**Reproducibility:**

4: Could mostly reproduce the results, but there may be some variation because of sample variance or minor variations in their interpretation of the protocol or method.

**Reviewer Confidence:**

3: Pretty sure, but there's a chance I missed something. Although I have a good feel for this area in general, I did not carefully check the paper's details, e.g., the math, experimental design, or novelty.

---

> ### Author Rebuttal · Authors · 2023-08-28
>
> We thank reviewer iKVr for their valuable feedback and want to address the following remark:
>
> **With respect to its performance, the only regret is a lack of qualitative comparison with previous methods. Using diffusion models, how the ScanDL performs different from previous models? Where is it strong, and how it performs differently from human?**
>
> In order to examine whether ScanDL and the baselines show the same trend in predictive performance, i.e., whether ScanDL and the baselines have good predictions on the same sentences, or bad predictions respectively, we averaged, for each model, the NLD of the different scanpaths generated by different readers on the same sentence, resulting in one mean NLD for each sentence and model. We then computed the correlations between the mean sentence NLDs of ScanDL and each of the baselines. Significant correlations are marked with an asterisk:
>
>
> | Baseline    | Pearson corr. |
> |-------------|---------------|
> | Eyettention | 0.40*         |
> | Traindist   | 0.37*         |
> | E-Z Reader  | 0.37*         |
> | Swift       | 0.36*         |
> | Uniform     | 0.35*         |
>
>
> We observe positive correlations between ScanDL and all baseline models, indicating that there is a tendency of all models to perform well on the same sentences. However, these correlations are not extremely strong (0.4 for Eyettention, and 0.36 and 0.37 for the two cognitive models), which, in turn, indicates that there are indeed differences between how well ScanDL and the baselines cope with certain reading patterns.
>
> To investigate in more detail what reading patterns ScanDL and the baselines struggle with or perform well on, we performed the following analyses: For ScanDL and each of the baselines, we computed the correlations between the NLD of a predicted scanpath and six reading measures characterizing the corresponding true scanpath from the test set. The reading measures are: the average probability of a word in the sentence to be the starting point of a regression (regression rate), the average number of fixations in a sentence (normalized fixation count), the average lengths of progressive and regressive saccades (prog. saccade length, regr. saccade length), the average probability of a word to be skipped (skips), and the average number of fixations on a word during the first-pass reading of the sentence (first-pass count).
> While for reviewer cEYa, the correlations we computed are on values aggregated over the readers and thus provide information about the model's reader-specific predictive ability, these instance-based correlations serve as indication of which specific scanpath properties are more or less difficult for a model. The results are depicted in the table below. Correlations with an absolute value greater than 0.3 are shown in bold font, statistically significant correlations with an asterisk:
>
>
>
> | Model       | Reading measure           | Pearson corr.  |
> |-------------|---------------------------|----------------|
> | ScanDL      | Normalized fixation count | **0.37***   |
> | ScanDL      | Prog. saccade length      | **0.35*** |
> | ScanDL      | Regression rate           | **0.32*** |
> | ScanDL      | Skips                     | 0.25*          |
> | ScanDL      | Regr. saccade length      | 0.21*          |
> | ScanDL      | First pass count          | 0.03           |
> | Eyettention | Prog. saccade length      | **0.32*** |
> | Eyettention | Normalized fixation count | 0.24* |
> | Eyettention | Skips                     | 0.19*          |
> | Eyettention | Regression rate           | 0.18 *         |
> | Eyettention | Regr. saccade length      | 0.18*          |
> | Eyettention | First pass count          | 0.01           |
> | E-Z Reader  | Normalized fixation count | **0.70*** |
> | E-Z Reader  | Regression rate           | **0.48*** |
> | E-Z Reader  | First pass count          | **0.47*** |
> | E-Z Reader  | Regr. saccade length      | 0.19*          |
> | E-Z Reader  | Prog. saccade length      | 0.02           |
> | E-Z Reader  | Skips                     | -0.20*         |
> | Swift       | Normalized fixation count | **0.68*** |
> | Swift       | Regression rate           | **0.46*** |
> | Swift       | First pass count          | **0.46*** |
> | Swift       | Regr. saccade length      | 0.15*          |
> | Swift       | Prog. saccade length      | 0.02           |
> | Swift       | Skips                     | -0.19*         |
> | Traindist   | Normalized fixation count | **0.33*** |
> | Traindist   | Regression rate           | 0.21*          |
> | Traindist   | Prog. saccade length      | 0.20*          |
> | Traindist   | Regr. saccade length      | 0.19*          |
> | Traindist   | First pass count          | 0.16*          |
> | Traindist   | Skips                     | 0.01           |
> | Uniform     | Normalized fixation count | **0.56*** |
> | Uniform     | First pass count          | **0.41*** |
> | Uniform     | Regression rate           | **0.37*** |
> | Uniform     | Regr. saccade length      | 0.23*          |
> | Uniform     | Prog. saccade length      | 0.03           |
> | Uniform     | Skips                     | -0.23*         |
>
>
> For all models, there is a significant positive correlation between the NLD and both the normalized fixation count and the regression rate. In other words, long scanpaths that exhibit a high number of fixations for a given sentence length, or scanpaths that exhibit a high number of regressions result in predictions with a higher NLD, meaning that they are more difficult to predict than shorter scanpaths (with a lower fixation count) and scanpaths with fewer regressions.
> Further, the patterns within types of models are similar, i.e., the two ML-based methods ScanDL and Eyettention exhibit the same correlation pattern, as well as the cognitive models E-Z Reader and Swift. Interestingly, both ScanDL and Eyettention have correlations close to zero with respect to the first-pass count, while both E-Z Reader and Swift display a strong correlation. Even more, the cognitive models have overall much stronger correlations than the ML-based models, with their respective correlations for the normalized fixation count being almost twice as high. They appear to be overfitting on specific reading patterns, while ScanDL and Eyettention both seem to generalize better across reading patterns.
>
> We have added a new subsection *Predictability Patterns Across Models* to the manuscript in which we present the above described analyses. We have also added a sentence to the Discussion discussing these results.
>
> Furthermore, we are currently conducting an analysis investigating the instance-based correlation between the NLD and linguistically-based sentence metrics, such as the number of named entities, information density, and syntactic complexity. This analysis will also be included in the above-mentioned subsection.

---

### Official Review · Reviewer_JBSW · 2023-08-04

**Soundness:** 3

**Excitement:**

4: Strong: This paper deepens the understanding of some phenomenon or lowers the barriers to an existing research direction.

**Paper Topic And Main Contributions:**

In this paper, a diffusion model is proposed to generate synthetic scanpaths given sentence texts. Build upon the DiffSeq model, the proposed model include scan path in the input representation learning process, where in the inference time the scan path is obtained through an iterative de-noising process. The proposed method shows sota performance when compared to baselines.

**Reasons To Accept:**

The paper is generally well written with clear motivation, strong performance backed with solid experiments. The analysis part is also interesting to read.

**Reasons To Reject:**

It is not clear how useful the generated scanpaths are in downstream tasks, but as the authors mentioned in the limitation section, this probably can be left for future work.

**Reproducibility:**

4: Could mostly reproduce the results, but there may be some variation because of sample variance or minor variations in their interpretation of the protocol or method.

**Reviewer Confidence:**

3: Pretty sure, but there's a chance I missed something. Although I have a good feel for this area in general, I did not carefully check the paper's details, e.g., the math, experimental design, or novelty.

---

> ### Author Rebuttal · Authors · 2023-08-28
>
> We thank reviewer JBSW for their helpful input and would like to address the following remark:
>
> **It is not clear how useful the generated scanpaths are in downstream tasks, but as the authors mentioned in the limitation section, this probably can be left for future work.**
>
> Reviewer JBSW is right in that not showing how augmenting language models with synthetic eye movements can improve their performance on NLP downstream tasks might be regarded as a limitation of the paper. Unfortunately, the space was limited and we had to decide which aspects we wanted to focus on. As previous research already demonstrates the usefulness of synthetic eye movements, such as Sood et al. (2020, "Improving natural language processing tasks with human gaze-guided neural attention"), we opted for focusing on the model itself. As ScanDL is a novel approach to synthesizing eye movements, we chose to perform a thorough evaluation of every aspect of the model, not only assessing its performance in a range of settings that differ in the extent to which the model is forced to generalize out-of-distribution, but also investigating its cognitive plausibility in a psycholinguistic analysis and comparing it to other models (both cognitive and ML-based).

---

### Official Review · Reviewer_cEYa · 2023-08-05

**Soundness:** 4

**Excitement:**

3: Ambivalent: It has merits (e.g., it reports state-of-the-art results, the idea is nice), but there are key weaknesses (e.g., it describes incremental work), and it can significantly benefit from another round of revision. However, I won't object to accepting it if my co-reviewers champion it.

**Paper Topic And Main Contributions:**

The paper addresses the problem of generating human-like scanpaths for a given sentence. It proposes a multi-modal discrete sequence-oto-sequence diffusion model for generating the scanpaths and achieves state-of-the-art results on various within and across-dataset settings. Ablation results emphasize the importance of sentence conditioning, bert and positional embeddings. A psycholinguistic analysis of the generated scanpaths is also presented showing human-like reading ability of their model's outputs.

**Questions For The Authors:**

See above for questions.

**Reasons To Accept:**

1. The addressed problem is interesting and the proposed model has merits as shown by various ablation and evaluations under different settings (new reader, new sentence, new reader/sentence).
2. The presented psycholinguistics analysis is an interesting addition to the work providing insights into its alignment with the cognitive aspects of human reading behaviour.
3. The paper is well-written and easy to understand.

**Reasons To Reject:**

1. The model learns scanpath predictions for an average reader. It is not clear how diverse are the readers in the train set? For eg. how aligned are the scanpaths for readers who are on sides of the spectrum of the readers in the dataset? Although the model is able to generalize to new readers, it is not clear how different are the reading patterns of the readers (for same sentence experiments) in the train and test sets and to what extent the model is able to capture the differences.
2. What does multi-modality mean in this case? It seems like scanpaths are also considered as text while modeling,
3. Scanpaths are required during training for the x_idx embedding, it is not clear how that is initialized during inference. how is the length of the scanpath decided when it is sampled to concatenate with EMD(x)?

**Reproducibility:**

4: Could mostly reproduce the results, but there may be some variation because of sample variance or minor variations in their interpretation of the protocol or method.

**Reviewer Confidence:**

3: Pretty sure, but there's a chance I missed something. Although I have a good feel for this area in general, I did not carefully check the paper's details, e.g., the math, experimental design, or novelty.

---

> ### Author Rebuttal · Authors · 2023-08-28
>
> We thank Reviewer cEYa for their careful reading of the paper and helpful feedback and address their concerns as follows:
> <br>
>
> **1. The model learns scanpath predictions for an average reader. It is not clear *how diverse are the readers in the train set?* For eg. *how aligned are the scanpaths for readers who are on sides of the spectrum of the readers in the dataset?* Although the model is able to generalize to new readers, it is not clear *how different are the reading patterns of the readers* (for same sentence experiments) *in the train and test sets and to what extent the model is able to capture the differences*.**
>
> To answer the *question of the diversity of readers in the train set*, we would like to first provide some demographic information: there are 69 L1 English speakers in CELER (Berzak et al., 2022), of which 38 are female and 28 are male. The mean age is 26.3 $\pm$ 6.7 years. Berzak et al. (2022) have recruited the participants via a variety of sources, such as mailing lists, advertisements on social media, or message boards. We have now included these demographic statistics in Section B of the Appendix.
>
> Further descriptive statistics taken from the original CELER publication on mean reading measure values can be seen in the table below. They describe the mean scanpath properties and are aggregated over readers and over sentences. We have also added this table to Section B of the Appendix.
>
> | Measure                      | Value           |
> |------------------------------|-----------------|
> | Number of fixations per word | 1.3 $\pm$ 0.1   |
> | Saccade length in chars      | 8.5 $\pm$ 0.4   |
> | Skip rate                    | 0.36 $\pm$ 0.02 |
> | Regression rate              | 0.24 $\pm$ 0.01 |
>
>
> However, the values in this table are aggregated over all L1 participants. To address both *the question of reader diversity* in more detail as well as *the question on how well ScanDL manages to capture the reading patterns of different readers, especially of readers that are towards the edge of the spectrum*, we have conducted an analysis to quantify the general reading behavior of each reader: we first computed a set of commonly used reading measures averaged over all scanpaths of a reader. The reading measures are: the average probability of a word in the sentence to be the starting point of a regression (regression rate), the average number of fixations in a sentence (normalized fixation count), the average lengths of progressive and regressive saccades (prog. saccade length, regr. saccade length), the average probability of a word to be skipped (skips), and the average number of fixations on a word during the first-pass reading of the sentence (first-pass count). We then computed the mean NLD for each reader between the reader's true scanpaths and the scanpaths predicted by the model, and then correlated the mean reader NLD with each per-reader mean reading measure. The reading measures and the correlations can be seen in the following table. Correlations greater than 0.3 are in bold font, statistically significant correlations are marked with an asterisk:
>
>
>
>
> | Reading measure           | Pearson corr. w. NLD |
> |---------------------------|----------------|
> | Regression rate           | **0.56*** |
> | Normalized fixation count | **0.48*** |
> | Progr. saccade length     | **0.45*** |
> | Regr. saccade length      | **0.30**  |
> | Skips                     | 0.26           |
> | First pass count          | 0.01           |
>
>
> The  correlation of 0.56 between the NLD and a reader's average regression rate indicates that the model has more difficulties predicting the reading behavior of readers who show a large proportion of regressions than the reading patterns of readers that only regress rarely. Along the same lines, we observe a positive correlation between a reader's average normalized fixation count on a sentence and NLD: long scanpaths with many fixations (normalized by sentence length) are more difficult for the model to predict than short scanpaths. Looking at progressive and regressive saccades, the positive correlation between a reader's average saccade length and the mean NLD for that reader suggests that it is more difficult for the model to predict the reading behavior of readers that tend to make long saccades (both progressive and regressive).
>
> While the analysis above examines how well the model captures different reading patterns, we would also like to address reviewer cEYa's question on *the differences in reading patterns* in more detail. While we did perform 5-fold cross-validation to account for the diversity of readers between the train and test set and also performed an across-dataset evaluation to assess the model's ability to generalize to unseen readers and sentences, we have not specifically described the diversity within each set. To that end, for the above-mentioned reading measures, we computed the overall mean and standard deviation for both the predicted scanpaths and the true scanpaths for the CELER dataset (Berzak et al., 2022) so as to quantify the overall reading pattern variability and whether or not the model manages to emulate this variability. We are planning to perform the same analysis for the ZuCo dataset (Hollenstein et al., 2018) as well to be able to also compare how well the model manages to capture the diversity in a completely different dataset. Results can be see in the table below.
>
>
>
>
> | Reading measure           | mean $\pm$ sd (true scanpaths)  | mean $\pm$ sd   (predicted scanpaths)   |
> |---------------------------|-----------------|---------------------|
> | Regression rate           | 0.15 $\pm$ 0.10 | 0.08 $\pm$ 0.08     |
> | Normalized fixation count | 1.22 $\pm$ 0.48 | 1.01 $\pm$ 0.28     |
> | Progr. saccade length     | 2.01 $\pm$ 0.59 | 1.72 $\pm$ 0.47     |
> | Regr. saccade length      | 2.54 $\pm$ 2.07 | 1.73 $\pm$ 2.23     |
> | Skips                     | 0.30 $\pm$ 0.13 | 0.28 $\pm$ 0.12     |
> | First pass count          | 0.90 $\pm$ 0.19 | 0.89 $\pm$ 0.19
>
>
> We observe that the scanpaths generated by ScanDL are similar in diversity to the true scanpaths. Not only is ScanDL's mean value of each reading measure close to the true data, but, crucially, ScanDL also reproduces the variability in the scanpaths, underlining the model's ability to emulate the reader diversity: for instance, the variability with respect to regressive saccades is big in both the true scanpaths as well as the predicted ones, and it is small for both true and predicted with regards to skips and first pass counts. The only minor difference in variability occurs with respect to the normalized fixation count, with ScanDL depicting less overall variability in the number of fixations it produces.
>
> We have added a new subsection  *Investigation of Reading Pattern Predictability* to the manuscript where we present the analyses summarized above. Moreover, we now discuss the above described findings in the Discussion section.
>
>
>
>
> **2. What does multi-modality mean in this case? It seems like scanpaths are also considered as text while modeling**
>
> The term *multi-modal* can be understood on two different levels, so we thank reviewer cEYa for making us aware of possible misunderstandings. We used the term *multi-modal* on a merely conceptual level, i.e., to describe the difference between the two kinds of data: the eye movements and the textual data, which are of different modalities.
> However, on a technical level, as pointed out by the reviewer, we have erased the multi-modality by modeling the scanpaths as textual data. In order to avoid confusion, we have removed the term *multi-modal* in the abstract and the manuscript in lines 21, 88, 94, and 611, as well as from the keywords.
>
>
>
>
> **3. Scanpaths are required during training for the x\_idx embedding, it is not clear how that is initialized during inference. how is the length of the scanpath decided when it is sampled to concatenate with EMB(x)**
>
> We are grateful to reviewer cEYa for making us aware that the model input at inference time needs to be more clearly stated in order to avoid confusion.
>
> At inference time, the word index embedding $Emb\_{idx}(\mathbf{x}\_{idx})$ is replaced by Gaussian noise, i.e., we initialize it by means of sampling noise from a Standard Normal Distribution. This noise is added up with the position embedding and the BERT word embedding.
> For better clarity, we have thus changed the corresponding part (lines 333ff. in Section 4.4) in the manuscript as follows:
>
> * *original*:  "We initialize the scanpath as noise sampled from a standard normal distribution $ \mathbf{z}\_0^{\mathbf{f}} \sim \mathcal{N}(0, \mathbb{I})$, and concatenate it with $ Emb(\mathbf{x^w}) $ to obtain the model input $ \mathbf{z}\_{\tilde{T}} $."
> * *new*: "We replace the word index embedding of the scanpath $Emb_{idx}(\mathbf{x^f}\_{idx})$
> with Gaussian noise, initializing it as
> $\tilde{\mathbf{x}}^{\mathbf{f}}\_{idx} \sim \mathcal{N}(0, \mathbb{I})$.
> We then concatenate the new embedding
> $\tilde{Emb}(\mathbf{x^f}) = \tilde{\mathbf{x}}^{\mathbf{f}}\_{idx} + Emb\_{bert}(\mathbf{x^f}\_{bert}) + Emb\_{pos}(\mathbf{x^f}\_{pos})$
>  with $Emb(\mathbf{x^w})$ to obtain the model input $\mathbf{z}_{\tilde{T}}$"
>
> Correspondingly, we have also changed Figure 3 (model architecture) to visualize unambiguously that only the word index embedding is noised and only afterwards is it added up with the position embedding and the BERT word embedding.
>
> Relatedly, we realized that  we had  not provided details on how the length of a scanpath is determined. Hence, we have now added a clarifying sentence in Section 4.4 after line 350:"As the model learns to predict both the scanpath as well as PAD tokens up to the maximum length, we obtain the scanpaths by removing the model's predicted PAD tokens from the prediction."

---

### Meta-Review · Area_Chair_igTX · 2023-09-02

**Recommendation:** 4
**Confidence:** 5

**Metareview:**

This paper introduces a model for generating synthetic eye movements on text using a discrete diffusion model with adjustments that surpass human performance. While the primary contribution is methodological, the experimental evidence supports the paper's main claim. This work is anticipated to have broad applications across various disciplines due to the challenge of obtaining real gaze data. Synthetic eye movements can benefit cognitive research in areas like readability, grammaticality, sentiment analysis, and more. The improvements over previous state-of-the-art methods are substantial, potentially establishing this approach as a new standard for the task. Both AC and 3 reviewers acknowledged the contribution of this paper.

However, one notable concern is the absence of qualitative comparisons with previous methods. Partly due to the novelty of this research. While the paper demonstrates the superior performance of the ScanDL model, it lacks qualitative insights into how it differs from earlier techniques that also employ diffusion models. Understanding the strengths and differences in performance compared to human eye movements could enhance the paper's arguments for adopting a diffusion model for this task.

In summary, the paper's methodological contribution is significant, establishing a new state-of-the-art in generating human-like eye movements. The model's performance is well-supported by experiments, and the paper is well-written with clear motivation and interesting analysis. Additionally, the psycholinguistic analysis adds valuable insights into the alignment of the model's outputs with human reading behavior. In my mind, the paper could benefit from further exploration of the diversity of readers in the training set and a clearer explanation of how scanpaths are handled during inference. Despite some minor concerns exists, the detailed rebuttal addressed most of the problems raised by reviewers.

Consider the original paper, the reviewer’s comments, and the rebuttal, as the area chair, would like to to recommend acceptance of the paper and invite the author to take an oral presentation in EMNLP.

---

### Decision · Program_Chairs · 2023-10-07

**Decision:**

Accept-Main

**Comment:**

This paper introduces a model for generating synthetic eye movements on text using a discrete diffusion model with adjustments that surpass human performance. While the primary contribution is methodological, the experimental evidence supports the paper's main claim. This work is anticipated to have broad applications across various disciplines due to the challenge of obtaining real gaze data. Synthetic eye movements can benefit cognitive research in areas like readability, grammaticality, sentiment analysis, and more. The improvements over previous state-of-the-art methods are substantial, potentially establishing this approach as a new standard for the task. Both AC and 3 reviewers acknowledged the contribution of this paper.

However, one notable concern is the absence of qualitative comparisons with previous methods. Partly due to the novelty of this research. While the paper demonstrates the superior performance of the ScanDL model, it lacks qualitative insights into how it differs from earlier techniques that also employ diffusion models. Understanding the strengths and differences in performance compared to human eye movements could enhance the paper's arguments for adopting a diffusion model for this task.

In summary, the paper's methodological contribution is significant, establishing a new state-of-the-art in generating human-like eye movements. The model's performance is well-supported by experiments, and the paper is well-written with clear motivation and interesting analysis. Additionally, the psycholinguistic analysis adds valuable insights into the alignment of the model's outputs with human reading behavior. In my mind, the paper could benefit from further exploration of the diversity of readers in the training set and a clearer explanation of how scanpaths are handled during inference. Despite some minor concerns exists, the detailed rebuttal addressed most of the problems raised by reviewers.

Consider the original paper, the reviewer’s comments, and the rebuttal, as the area chair, would like to to recommend acceptance of the paper and invite the author to take an oral presentation in EMNLP.